# OV-PARTS: Towards Open-Vocabulary Part Segmentation

**Meng Wei** [1,2]   **Xiaoyu Yue** [3]   **Wenwei Zhang** [1]   **Shu Kong** [4,5]
**Xihui Liu** [2]   **Jiangmiao Pang** [1*]

[1]Shanghai AI Laboratory   [2]The University of Hong Kong
[3]The University of Sydney   [4]University of Macau   [5]Texas A&M University

mengwei.kelly@connect.hku.hk   yuexiaoyu002@gmail.com   skong@um.edu.mo
xihuiliu@eee.hku.hk   {zhangwenwei, pangjiangmiao}@pjlab.org.cn

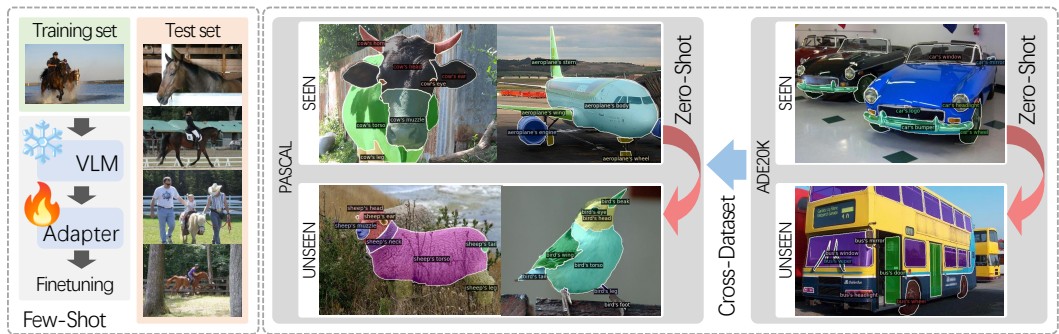

Figure 1: An illustration of the Generalized Zero-Shot Setting (red arrow), Cross-Dataset Setting (blue arrow) and Few-Shot Setting with examples from Pascal-Part-116 and ADE20K-234.

## Abstract

Segmenting and recognizing diverse object parts is a crucial ability in applications spanning various computer vision and robotic tasks. While significant progress has been made in object-level Open-Vocabulary Semantic Segmentation (OVSS), *i.e.*, segmenting objects with arbitrary text, the corresponding part-level research poses additional challenges. Firstly, part segmentation inherently involves intricate boundaries, while limited annotated data compounds the challenge. Secondly, part segmentation introduces an open granularity challenge due to the diverse and often ambiguous definitions of parts in the open world. Furthermore, the large-scale vision and language models, which play a key role in the open vocabulary setting, struggle to recognize parts as effectively as objects. To comprehensively investigate and tackle these challenges, we propose an **O**pen-**V**ocabulary **Part S**egmentation (**OV-PARTS**) benchmark. OV-PARTS includes refined versions of two publicly available datasets: Pascal-Part-116 and ADE20K-Part-234. And it covers three specific tasks: *Generalized Zero-Shot Part Segmentation*, *Cross-Dataset Part Segmentation*, and *Few-Shot Part Segmentation*, providing insights into analogical reasoning, open granularity and few-shot adapting abilities of models. Moreover, we analyze and adapt two prevailing paradigms of existing object-level OVSS methods for OV-PARTS. Extensive experimental analysis is conducted to inspire future research in leveraging foundational models for OV-PARTS. The code and dataset are available at `https://github.com/OpenRobotLab/OV_PARTS`.

---

*Corresponding Author.

37th Conference on Neural Information Processing Systems (NeurIPS 2023) Track on Datasets and Benchmarks.

# 1 Introduction

The ability to identify and reason about object parts is crucial for a wide range of human activities. For instance, when preparing a meal, we rely on specific parts of utensils such as the blade of a knife for slicing and the handle of a spatula for stirring. Hence, developing a vision system capable of part-level object segmentation is crucial and offers substantial benefits across applications in vision and robotics such as image editing [14, 19], object manipulation [27] *etc*. Despite dedicated efforts in annotating fine-grained parts by previous works [4, 34], the complex nature and diverse granularity of object parts make it hard to create a comprehensive closed category set. Recently, the research on Open Vocabulary Semantic Segmentation (OVSS) [22, 9, 7, 13, 33, 32] extends the image-text alignment ability of large-scale Vision-Language Models (VLM) like CLIP [26] to pixel-level prediction, which shows remarkable performance in open vocabulary object segmentation.

Despite the satisfying performance on object-level OVSS, part-level OVSS raises additional challenges. Parts exhibit complex structures, often with more intricate boundaries and appearance variations than objects. However, the available labeled data for training part segmentation models is significantly limited compared to that of objects. Moreover, in the open-vocabulary setting, there are further challenges to overcome. Firstly, while objects are typically well-defined entities with strict boundaries, the granularity of parts can be flexible, posing the extra open granularity challenge that rarely occurs in object-level OVSS. Secondly, the widely used large-scale VLM are mainly pretrained on natural image-text pairs which are inherently biased to object words. Hence, their ability to recognize object parts can be less proficient. This disparity is reflected in the less discriminative class activation maps of CLIP [26] for parts as opposed to objects, as illustrated in Figure 2 (a).

Considering these challenges, we propose to break down the complex part-level OVSS problem into specific subtasks. One key observation is, common objects often exhibit shared characteristics in terms of parts. For example, furniture like chair, table and sofa often have shared parts such as legs, seats, and backrests. Humans can categorize a new object based on its part descriptions related to a known object. Similarly, we can leverage this **analogical reasoning** ability to improve data efficiency in part-level OVSS. Hence, we propose a *Generalized Zero-Shot Part Segmentation* setting, focusing on assessing the transferability of part segmentation from seen objects to related unseen objects. Additionally, we extend the related *Cross-Dataset Segmentation* setting, commonly used in object-level OVSS, to part-level OVSS. This setting further emphasizes the **open granularity** challenge, due to the varying part vocabularies and granularity levels across datasets, as shown in Figure 1. Finally, concerning the weaker transferability of large-scale foundation models for the part, we further expect a *Few-Shot Part Segmentation* setting to enable **fast adaptation** of the foundation models to part-level OVSS. The entire benchmark, named **O**pen-**V**ocabulary **Part S**egmentation (**OV-PARTS**), is further supplemented with carefully cleaned and reorganized versions of Pascal-Part [4] and ADE20K-Part [34], namely Pascal-Part-116 and ADE20K-Part-234.

Additionally, we design strong baselines for OV-PARTS based on two paradigms (*i.e.*, two-stage and one-stage) of existing object-level OVSS methods. The first paradigm [9, 33, 32] designs a two-stage approach that decouples the segmentation and open vocabulary classification abilities. Applying this paradigm to OV-PARTS involves training a class-agnostic part proposal model followed by using CLIP for part region classification. However, it turns out that treating object part as independent classes like objects lead to suboptimal performance. Indeed, even humans struggle to distinguish between "cow's leg" and "sheep's leg" if only the region of their legs is shown. To mitigate this problem, we propose two improvements: (1) An **Object Mask Prompt** strategy which introduces the object-awareness to the first part proposal stage. (2) A **Compositional Prompt Tuning** strategy which not only enhances object awareness in the second stage but also shifts CLIP's attention from objects to object parts. This two-stage paradigm offers the advantage of combining class-agnostic part parsing models [15, 25] and various finetune methods to adapt foundation models for classification [38, 37, 2]. However, using a class-agnostic part proposal model has limitations. It is trained on pre-defined parts, which is not applicable to the open granularity scenario. Moreover, the mask proposal model is inclined to overfit the training data, leading to reduced zero-shot generalization ability.

The other line of works [7, 22] follows a one-stage paradigm that trains a unified open-vocabulary segmentation model based on CLIP, eliminating the missing object context problem. As for the open granularity ability, we experimentally find that CATSeg [7] and CLIPSeg [22], which are pretrained on COCO-Stuff[1] and PhraseCut [31] respectively, can already achieve the first-level granular generalization from object to some simple parts as shown in Figure 2 (b). Motivated by this,

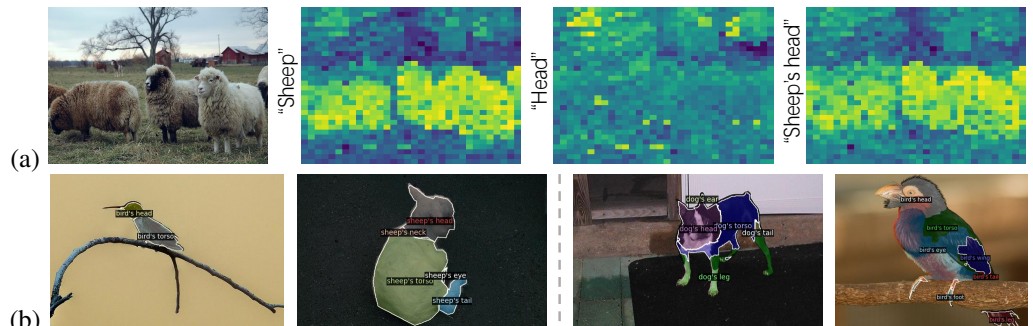

Figure 2: (a) Class activation maps of CLIP. The object prompt "Sheep" can activate its area while the part prompt "Head" fails. When using "Sheep's Head" as prompt, the activation is still biased to "Sheep". (b) Without finetuning on part datasets, CATSeg outputs rough part masks of "Bird" and "Sheep" (**Left**) and CLIPSeg produces finer granular part masks of "Bird" and "Dog" (**Right**.).

we design some one-stage baselines by finetuning specific modules of CLIPSeg and also investigate parameter-efficient finetuning strategies [12] for OV-PARTS. However, the segmentation ability of one-stage baselines is weaker than the two-stage baselines, mainly limited by the frozen CLIP visual encoder which is pretrained without sufficient segmentation data.

In summary, the two-stage and one-stage baselines exhibit complementary strengths and weaknesses. But there are no currently clear solutions about how to achieve both high-quality segmentation and strong open vocabulary and granularity generalization ability in OV-PARTS. Hence, there remains much to uncover in addressing OV-PARTS, particularly in unlocking the full potential of large foundation models in vision, language and multi-modality, which can be the value of the benchmark.

## 2   Related Work

**Open Vocabulary Semantic Segmentation.** Open vocabulary semantic segmentation has recently achieved significant progress with the help of large-scale Vision-Language Models (VLM) like CLIP [26], which excel at recognizing objects in images. One line of research [9, 33, 32] combines powerful segmentation models like MaskFormer [5] with CLIP in a two-stage way. In the first stage, MaskFormer [5] produces class-agnostic object mask proposals. In the second stage, CLIP [26] classifies the image regions of these proposals. ODISE [32] further used a pretrained diffusion model to enhance the proposal stage. Another one-stage approach focuses on extending CLIP [26] to pixel-level prediction [7, 22]. They mainly train a pixel decoder on top of the CLIP image encoder. CLIPSeg [22] adds a lightweight transformer-based pixel decoder with a FiLM[10] module to fuse the multi-modality features. CATSeg [7] designs a spatial and class aggregation network with multi-modality guidance features for effective open vocabulary pixel classification.

**Part Segmentation.** Fine-grained part segmentation has been actively studied in the literature [17, 8, 39, 24, 23, 29, 25, 28]. Most of these methods [17, 8, 39, 24, 23] adopt a supervised closed-set setting. Tang et al. [29] designs a supervised language-driven segmentation model which allows interactive whole-to-part segmentation. Recently, Pan et al. [25] proposes an open-world part segmentation setting that only focuses on the class-agnostic part mask generation which is not language-driven. The concurrent work Sun et al. [28] tackles the open vocabulary part segmentation task with a cross-dataset setting. However, the big performance gap compared to open vocabulary object segmentation hasn't been well studied. Also, their proposed method emphasizes building semantic correspondences based on visual features. But the role of language in the open vocabulary setting as well as the potential of vision language models has not been adequately discussed. Moreover, some existing open-vocabulary object detection models [16, 18, 20] can also detect parts with bounding boxes. Because they are trained on crowd-sourced datasets in which the texts can also include part words. However, pixel-level understanding is more conforming to the nature of part which is ambiguous, multi-granular and has intricate boundaries. For example, with bounding boxes, it'll be hard to identify a little bird's parts accurately.

# 3 Dataset and Benchmark Details

In this section, we first introduce the two proposed datasets Pascal-Part-116 and ADE20K-Part-234 in section 3.1. Then we elaborate on the three task settings designed for **OV-PARTS** in section 3.2. The exhaustive list of object part classes, the specific data splits and the distribution of part scales and numbers of Pascal-Part-116 and ADE20K-Part-234 are left to the supplementary material.

## 3.1 Datasets

**Pascal-Part-116.** Pascal-Part dataset [4] is an extension of the PASCAL VOC 2010 dataset [11], further annotating objects' part masks. Some categories such as "cow", are annotated with a comprehensive list of parts, while others like "chair", "boat", and "dining table" only provide silhouette annotations. Moreover, the part definition includes directional terms like "left," "right," "front," "back," "upper," and "lower", such as "cow's left front lower leg". However, in an open-vocabulary setting, it's unnecessary to discern between the semantics of labels such as "cow's left front lower leg" and "cow's right back lower leg". They can not only create a bottleneck in part segmentation but also cause overfitting which hinders effective language-driven generalization. Hence, we have manually merged some of the over-segmentation parts to create a more practical version. Our revised Pascal-Part dataset [4] includes a total of 116 object part classes across 17 object classes, which is the most extensive set among various versions of Pascal-Part dataset[30, 21, 3, 24].

**ADE20K-Part-234.** The ADE20K dataset [34] provides open-ended annotations of 847 objects and 1000+ parts, following the WordNet hierarchy. It covers a broad range of scenes, including indoor spaces such as "bedrooms", and outdoor spaces like "streetscapes". However, the part annotations in ADE20K are extremely sparse and incomplete (less than $15\%$ object instances have part annotations), which poses significant challenges for both training models and evaluating their performance. Despite attempts to reorganize the dataset [23, 29], either the revised versions have not been publicly released or they still contain considerable noise. To get a clean version, we started with the widely used SceneParse150 [35] subset and only keep the objects which have more than one frequently annotated part (over 100 occurrences) and then filter the rare parts (less than 10 occurrences). Moreover, we manually merge some duplicated parts such as "chair arm" and "chair armrest", "table stretcher" and "table h-stretcher" as well as the over-segmentation parts. The resulting subset consists of 44 objects and 234 parts, providing a cleaner dataset for improved analysis and evaluation.

## 3.2 Benchmark Tasks

There are two primary challenges : (1) The available pixel-level part data is limited. (2) Pretrained features from large-scale VLM exhibits weaker transferability to parts. To evaluate the OV-PARTS models comprehensively, we have designed three task settings: Generalized Zero-Shot Part Segmentation, Cross-Dataset Part Segmentation, and Few-Shot Part Segmentation.

**Generalized Zero-Shot Part Segmentation.** Considering the limited ability of VLMs to recognize parts, this task aims to assess the model's analogical reasoning ability, which is designed by selecting novel objects that possess related parts to the base objects, rather than being completely irrelevant. *Data Split*. To split the object classes in each dataset, we group the object classes into higher-level categories (e.g. Animals, Vehicles) based on their shared attributes. Within each hyper-category, we split the objects into base and novel classes (74/42 for Pascal-Part-116 and 176/58 for ADE20K-Part-234). The unseen objects in the training set are set to the background. In this way, a novel object part class may be novel at the object level (*e.g.*, "dog's head" is a novel class while "cat's head" is a base class) or both at the object level and the part level (*e.g.*, "bird's beak"). The complete base and novel class set can be found in the supplementary material. *Evaluation Protocol*. Following previous OVSS methods [33, 9], we first calculate the mean class-wise Intersection over Union (mIoU) on both base and novel classes. Then, to provide a balanced assessment of the model's performance across both base and novel classes in this setting, we use harmonic mean IoU (hIoU) as the primary metric.

**Cross-Dataset Part Segmentation.** In object-level OVSS, the cross-dataset setting evaluates the model's ability to handle variations in data distribution and novel object vocabulary. While in OV-PARTS, considering diverse part definitions, the model further needs to generalize between different annotation granularity levels in addition to different vocabularies. For example, the part set of "car" is

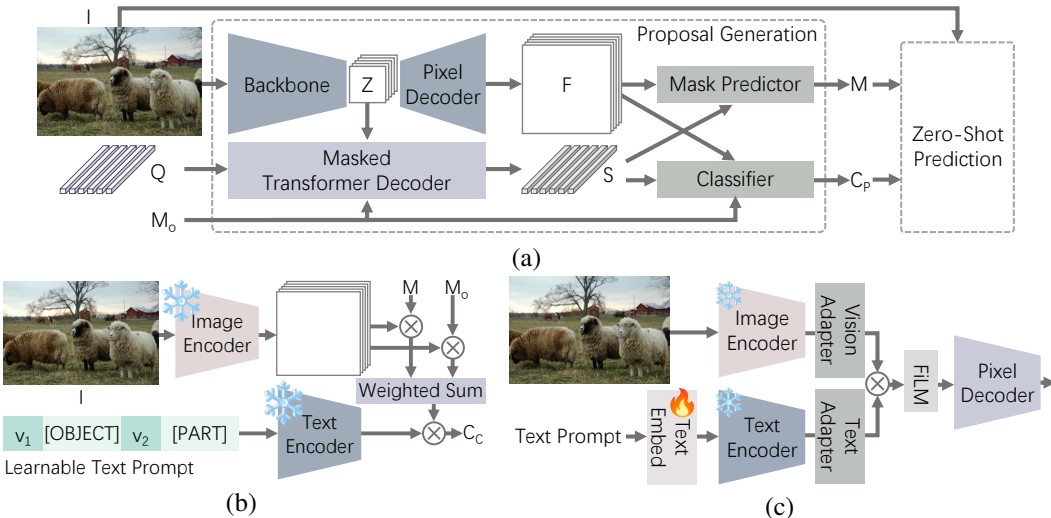

Figure 3: The overall frameworks of the representative two-stage (ZSseg+) and one-stage (CLIPSeg) baselines. (a) is the modified proposal generation stage of ZSseg. (b) shows the design of Compositional Prompt Tuning based on CoOp. (c) is the CLIPSeg model architecture which also indicates the modules that can be finetuned. CLIP is fixed in both frameworks.

["wheel", "headlight", "license plate", "mirror", "door", "window", "side", "front", "back", "roof"] in Pascal-Part-116 while is ["bumper", "door", "headlight", "hood", "license plate", "logo", "mirror", "wheel", "window", "wiper"] in ADE20K-Part-234 which has a finer granularity.

*Data Split.* Overall, ADE20K-234 covers a more diverse set of objects than Pascal-Part-116. We train models on the full training set of ADE20K-234 and evaluate them on Pascal-Part-116's testing set.

*Evaluation Protocol.* We reported mIoU on both the source dataset (ADE20K-Part-234) and target dataset (Pascal-Part-116).

**Few-Shot Part Segmentation.** Indeed, there's still a big performance gap between OV-PARTS models compared to the object-level OVSS models, mainly due to the inherent task difficulty and data limitation. Inspired by recent advances in foundation models, we design this few-shot part segmentation setting mainly to explore the strategies to adapt large-scale foundation models effectively for OV-PARTS and investigate the feasibility of utilizing object-level data to achieve parameter-efficient training and data efficiency.

*Data Split.* We sample 16 images for each object class. Since there may be multiple objects in an image or each object may not have exhaustive part annotations, the exact number of sampled shots for each object part class may be slightly over or below 16.

*Evaluation Protocol.* We use mIoU as the main metric.

# 4   Methodology

Existing object-level Open Vocabulary Semantic Segmentation (OVSS) methods can be categorized into the two-stage and one-stage paradigms. These paradigms differ in the way to apply the CLIP model to the segmentation process. In this section, we will give a brief overview of the framework for each paradigm. We then introduce our modifications with insights into the baselines.

## 4.1   Two-Stage Baseline

For the two-stage framework (Figure 3 (a)), we select ZSseg [33] as the baseline, which includes a proposal generation stage and a zero-shot prediction stage. The proposal generation stage mainly includes a MaskFormer [5]. Given an input image $\mathbf{I}$, a backbone network extracts the image features $\mathbf{Z}$. Then a pixel decoder is used to get the mask features $\mathbf{F}$ and a query-based transformer decoder is used to get the query features $\mathbf{S}$ with learnable queries $\mathbf{Q}$. Finally, $\mathbf{S}$ is fed into a mask predictor to get the class-agnostic masks $\mathbf{M}$ by computing the dot product of $S$ and $F$ and a fixed classifier

(initialized from CLIP text embeddings) to compute the classification score $C_P$:

$$S = TransformerDecoder(Q; Z) \tag{1}$$

$$M = MaskPredictor(S; F), C_P = Classifier(S) \tag{2}$$

In the zero-shot prediction stage, the mask proposals $\mathbf{M}$ are used to crop the original image $\mathbf{I}$. The resulting masked image is then fed into the CLIP model for zero-shot class prediction. The final classification scores are obtained through an ensemble of both stages:

$$C_C = CLIP(I; M), C = Ensemble(C_P; C_C) \tag{3}$$

Indeed, directly transferring the zsseg method from object-level data to part-level data has limitations and can lead to suboptimal performance. Unlike objects, which exist independently, parts are always contextually dependent on their respective objects. Classifying part regions without considering their object context is challenging. To address this, we propose two approaches to incorporate object context into both stages:

**Object Mask Prompt.** In the first stage, we propose to incorporate object masks as a mask prompt which can be obtained in a more cost-effective manner. This decoupling of object masks from the part segmentation brings object awareness to the model and helps resolve the difficulty. Furthermore, it enables interactive segmentation applications such as image editing.

In detail, as shown in Figure 3 (a), we modify the transformer decoder by replacing the cross-attention layers with mask-guided cross-attention layers [6]. This ensures that the cross-attention mechanism is computed only within the regions corresponding to objects, resulting in more accurate and context-aware part proposals. Equation (1) thus becomes:

$$S = MaskedTransformerDecoder(Q; Z; M_o) \tag{4}$$

where $M_o$ indicates the object mask. We further find that the proposals of parts are very noisy compared to objects. Hence, we design a mask-denoising technique during testing. Instead of directly using $\mathbf{M}$ for the second stage, we aggregate $\mathbf{M} \in \mathbb{R}^{q \times h \times w}$ and $C_P \in \mathbb{R}^{q \times c}$ to get a refined set of part mask proposals:

$$C_C = CLIP(I; Binary(\sum_q (C_P \times M) \in \mathbb{R}^{c \times h \times w})) \tag{5}$$

Here, $q, c, h$ and $w$ represent the number of learnable queries, the number of classes, the image height and the image width respectively. In the $Binary$ operation, we process the semantic mask into $c$ binary masks.

**Compositional Prompt Tuning.** In the second stage, to ensure that CLIP is not confused when presented with only partial regions of objects, we adapt CLIP's vision encoder following MaskCLIP [36] to obtain dense image embedding instead of the global image embedding. Then we use mask pooling to get CLIP visual features for both the object mask and the part mask. Since we observed that objects tend to capture most of CLIP's attention, overshadowing the parts, we design a compositional prompt tuning method based on CoOp [38]. We add object tokens and part tokens for prompt tuning with a learnable fusion weight. The modified zero-shot prediction stage is shown in Figure 3 (b). We find that this prompt tuning strategy is effective with minimal data and training cost.

## 4.2 One-Stage Baseline

In the one-stage framework, we employ CLIPSeg [22] as our baseline. The overall framework of CLIPSeg is shown in Figure 3 (c). CLIPSeg adds a parameter-efficient three-layered transformer decoder to the original CLIP model for segmentation. It integrates visual features from the final layer of the visual encoder and text features of all object part prompts from the text encoder through the FiLM module, forming cross-modal input token embeddings for the decoder. Furthermore, the features extracted from the 3rd, 6th, and 9th layers of CLIP's visual encoder are projected and added to the intermediate features of the corresponding decoder layers. It's worth noting that, the visual features extracted from the frozen CLIP visual encoder first pass through the added visual adapter, which consists of a two-layered MLP, before reaching the decoder. CLIPSeg is originally designed to generate binary segmentation maps conditioned on either a visual prompt or a text prompt. So we replace the original binary segmentation head with a multi-class one and modify its loss for multi-class segmentation using only text prompts.

By directly applying CLIP to the pixel-level prediction, we find that CLIPSeg, trained on Phrase-Cut [31] with object phrases as text input, shows amazing zero-shot generalization to parts. From this finding and considering the data sparsity issue in part segmentation, our focus shifts to exploring parameter-efficient finetuning for the one-stage baseline. Finetuning the entire CLIP model has proven to be harmful to its generalization ability [36]. Hence we conduct experiments to finetune different combinations of the specific modules of CLIPSeg, including the text embedding layer in CLIP, the pixel decoder and the extra light-weight CLIP-Adapter modules [12] (Vision-Adapter and Text-Adaper are simple MLPs) as shown in Figure 3 (c).

## 5 Experiments

### 5.1 Experimental Setup

**Evaluation Setup for Three Task Settings**. Since the state-of-the-art object-level OVSS models are powerful, we design two evaluation settings to investigate the primary bottlenecks: (1) **Oracle-Obj Setting**: The ground-truth mask and class of objects are assumed to be known. (2) **Pred-Obj Setting**: The ground-truth mask and class of objects are not provided.
**Implementation Details.** The details of baselines and training are left to the supplementary material.

### 5.2 Results in Generalized Zero-Shot Part Segmentation

*Settings*. We reported results of both two-stage and one-stage baselines on Pascal-Part-116 and ADE20K-Part-234 datasets as shown in Table 1 and Table 2 respectively. Firstly, to provide a comprehensive comparison, we reported the results of the supervised baseline: **MaskFormer** [5] with ResNet-50 backbone on the seen and unseen classes. Then for the *two-stage baselines*, we reported the results of the original **ZSseg** [33] and our modified ZSseg (**ZSseg+**) with different finetuning methods for the second stage: Compositional Prompt Tuning based on CoOp [38] and CoCoOp [37], abbreviated to **CPT-CoOp** and **CPT-CoCoOp**. For the *one-stage baselines*, we examined two state-of-the-art methods: **CAT-Seg** [7] and **CLIP-Seg** [22]. We used the pretrained model of CATSeg trained on COCO-Stuff and CLIPSeg trained on PhraseCut. Then we reported their results without any finetuning, as well as the results when adopting various finetuning strategies. For the Pred-Obj Setting evaluation, we use ZSseg's pretrained object model on COCO-Stuff in the two-stage baselines while don't adopt separate object models in one-stage baselines.

*Two-Stage Baselines*. We evaluated the effects of the **Object Mask Prompt**, **Mask Denoise**, and **CPTCoOp** in ZSseg+ on Pascal-Part-116 as shown in Table 3. We gradually add each approach to **ZSseg** to measure its individual effect under the Oracle-Obj Setting. The results indicate that the

Table 1: Zero-shot performance of the two-stage and one-stage baselines on Pascal-Part-116.

| Model | Backbone | Finetuning | Oracle-Obj | | | Pred-Obj | | |
|---|---|---|---|---|---|---|---|---|
| | | | Seen | Unseen | Harmonic | Seen | Unseen | Harmonic |
| *Fully-Supervised Baseline* | | | | | | | | |
| MaskFormer [5] | ResNet-50 | ✗ | 55.28 | 52.14 | - | 53.07 | 47.82 | - |
| *Two-Stage Baselines* | | | | | | | | |
| ZSseg [33] | ResNet-50 | ✗ | 49.35 | 12.57 | 20.04 | 40.80 | 12.07 | 18.63 |
| ZSseg+ | ResNet-50 | CPTCoOp | 55.33 | 19.17 | 28.48 | 54.23 | 17.10 | 26.00 |
| ZSseg+ | ResNet-50 | CPTCoCoOp | 54.43 | 19.04 | 28.21 | 53.31 | 16.08 | 24.71 |
| ZSseg+ | ResNet-101c | CPTCoOp | **57.88** | **21.93** | **31.81** | **56.87** | **20.29** | **29.91** |
| *One-Stage Baselines* | | | | | | | | |
| CATSeg [7] | ResNet-101 & ViT-B/16 | ✗ | 14.89 | 10.29 | 12.17 | 13.65 | 7.73 | 9.87 |
| CATSeg [7] | ResNet-101 & ViT-B/16 | B+D | 43.97 | 26.11 | 32.76 | 41.65 | 26.08 | 32.07 |
| CLIPSeg [22] | ViT-B/16 | ✗ | 22.33 | 19.73 | 20.95 | 14.32 | 10.52 | 12.13 |
| CLIPSeg [22] | ViT-B/16 | VA+L+F+D | **48.68** | **27.37** | **35.04** | **44.57** | **27.79** | **34.24** |

Table 2: Zero-shot performance of the two-stage and one-stage baselines on ADE20K-Part-234.

| Model | Backbone | Finetuning | Oracle-Obj | | | Pred-Obj | | |
|---|---|---|---|---|---|---|---|---|
| | | | Seen | Unseen | Harmonic | Seen | Unseen | Harmonic |
| *Fully-Supervised Baseline* | | | | | | | | |
| MaskFormer [5] | ResNet-50 | ✗ | 46.25 | 47.86 | - | 35.52 | 16.56 | - |
| *Two-Stage Baselines* | | | | | | | | |
| ZSseg+ | ResNet-50 | CPTCoOp | 43.19 | **27.84** | **33.85** | 21.30 | **5.60** | **8.87** |
| ZSseg+ | ResNet-50 | CPTCoCoOp | 39.67 | 25.15 | 30.78 | **19.52** | 2.98 | 5.17 |
| ZSseg+ | ResNet-101c | CPTCoOp | **43.41** | 25.70 | 32.28 | **21.42** | 3.33 | 5.76 |
| *One-Stage Baselines* | | | | | | | | |
| CATSeg [7] | ResNet-101 & ViT-B/16 | ✗ | 11.49 | 8.56 | 9.81 | 6.30 | 3.79 | 4.73 |
| CATSeg [7] | ResNet-101 & ViT-B/16 | B+D | 31.40 | 25.77 | 28.31 | 20.23 | **8.27** | **11.74** |
| CLIPSeg [22] | ViT-B/16 | ✗ | 15.27 | 18.01 | 16.53 | 5.00 | 3.36 | 4.02 |
| CLIPSeg [22] | ViT-B/16 | VA+L+F+D | **38.96** | **29.65** | **33.67** | **24.8** | 6.24 | 9.98 |

Table 3: Ablations on proposed strategies in ZSseg+ by adding each module to ZSseg on Pascal-Part-116.

| Model | Oracle-Obj | | |
|---|---|---|---|
| | Seen | Unseen | Harmonic |
| ZSseg [33] | 49.35 | 12.57 | 20.04 |
| +Obj Mask Prompt | 48.00 | 13.89 | 21.54 |
| +Mask Denoise | 48.00 | 16.84 | 24.93 |
| +CPTCoOp | 55.33 | 19.17 | 28.48 |

Table 4: Cross-Dataset Performance. Models are trained on the source dataset ADE20K-Part-234 and tested on the target dataset Pascal-Part-116.

| Model | Source | | Target | |
|---|---|---|---|---|
| | Oracle | Pred | Oracle | Pred |
| CATSeg | 27.95 | 17.22 | 16.00 | **14.72** |
| VA+L+F | 35.01 | 21.74 | 16.18 | 11.70 |
| VA+L+F+D | **37.76** | **21.87** | **19.69** | 13.88 |

**Object Mask Prompt** improves the performance on unseen classes ($+1.32\%$), and adding Mask Denoise further improves it ($+2.95\%$). However, there is a slight performance drop ($-1.35\%$) on the seen classes. Moreover, finetuning the CLIP embeddings with **CPTCoOp** leads to significant performance gains on both the seen ($+7.33\%$) and unseen ($+2.33\%$) classes. Notably, the finetuning with CPTCoOp requires only 500 iterations and less than 128 samples per class.

*One-Stage baselines*. Both **CATSeg** and **CLIPSeg** employ a frozen CLIP visual encoder to extract image features. But **CATSeg** further uses an extra learnable visual backbone to guide the pixel decoder. In Table 1 and Table 2, we can see that the **CATSeg** and **CLIPSeg** models, pretrained on object datasets, already show impressive results. Notably, **CLIPSeg** even outperforms ZSseg+ with a ResNet-50 backbone on the unseen classes. The comparison between **CLIPSeg** and **CATSeg** shows that OVSS models trained with phrases of objects exhibit stronger generalization ability to parts than sole objects. Hence, we explore alternative finetuning strategies rather than training from scratch. We finetune three different components: language embedding layer in text encoder (**L**), FiLM (**F**) and Decoder (**D**) of **CLIPSeg** and two added lightweight modules: CLIP-Adapter [12] to the visual encoder (**VA**) and text encoder (**TA**). As shown in Table 5, we draw three key conclusions: (1) Multi-modality finetuning is better than single-modality (VA+L+FiLM surpasses VA+FiLM and L+FiLM).; (2) Finetuning language embedding performs better than the text adapter (VA+L+F surpasses VA+TA+F); (3) Parameter-efficient finetuning can effectively transfer the knowledge from large-scale foundation models and object-level datasets to part parsing. We can see that VA+L+F even outperforms finetuning the entire pixel decoder on the unseen classes.

The qualitative results, which present a comparison among **ZSseg+**, **CATSeg** and **CLIPSeg** on the unseen class "Bird" of Pascal-Part-116, as well as additional qualitative results on Pascal-Part-116 and ADE20K-Part-234, are available in Section C of the supplementary material.

## 5.3 Results in Few-Shot Part Segmentation

In the few-shot setting, the two-stage baselines have inherent limitations. Firstly, training the MaskFormer from scratch is prone to overfitting with limited data. Secondly, the class-agnostic proposal generation is unable to generalize from object to part, making it hard to make use of the

Table 5: Performance on finetuning various modules of CLIPSeg in both the zero-shot setting and few-shot setting on Pascal-Part-116 and ADE20K-Part-234.

| Model | Pascal-Part-116 | | | | | ADE20K-Part-234 | |
|---|---|---|---|---|---|---|---|
| Setting | Zero-Shot Setting (Oracle) | | | Few-Shot Setting | | Few-Shot Setting | |
| Finetuning | Seen | Unseen | Harmonic | Oracle | Pred | Oracle | Pred |
| ✗ | 22.33 | 19.73 | 20.95 | 21.58 | - | 15.38 | - |
| D | 44.65 | 26.03 | 32.89 | 29.86 | 27.16 | 24.01 | 12.96 |
| F | 31.34 | 21.45 | 25.46 | - | - | - | |
| L+F | 38.11 | 21.14 | 27.19 | 27.61 | 25.90 | 26.04 | 14.55 |
| TA+F | 33.13 | 21.70 | 26.23 | - | - | - | - |
| VA+F | 45.13 | 22.55 | 30.07 | 29.99 | 26.84 | 23.74 | 12.72 |
| VA+TA+F | 44.83 | 24.16 | 31.39 | - | - | - | - |
| VA+L+F | 47.04 | **27.69** | 34.85 | 31.34 | 27.67 | 28.67 | **17.32** |
| VA+L+F+D | **48.68** | 27.37 | **35.04** | **33.13** | **30.70** | **29.36** | 16.12 |

object-level data. We reported the results of finetuning various modules in CLIPSeg as shown in Table 5. We can draw similar conclusions regarding the different finetuning strategies as observed in the zero-shot setting results. Besides, we observe inconsistent performance trends on Pascal-Part-116 and ADE20K-Part-234: (1) Visual modality finetuning is superior on Pascal-Part-116 while it is inferior on ADE20K-Part-234 (**L+F** v.s. **VA+F**). (2) Finetuning decoder (**D**) even underperforms the single language modality (**L+F**) on ADE20K-Part-234, whereas the opposite trend is observed on Pascal-Part-116. (3) Finetuning **VA+L+F+D** on ADE20K-Part-234 brings $0.69\%$ gains to **VA+L+F** under the **Oracle-Obj Setting** but causes $1.2\%$ performance drops ($1.2\%$) under the **Pred-Obj Setting**, which is different from the Pascal-Part-116. The cause of the inconsistencies is that, ADE20K-Part-234 is more challenging than Pascal-Part-116 thus the transferability gap from object to part is larger. Finetuning more parameters may cause worse performance and is biased to parts compared to objects. Qualitative results are left to Section C of the supplementary material.

## 5.4 Results in Cross-Dataset Part Segmentation

We reported three baselines as shown in Table 4: **CATSeg** with backbone and decoder finetuned, CLIPSeg with Visual Adapter, Language Embedding and FiLM (**VA+L+F**) finetuned and further the decoder (**VA+L+F+D**) finetuned. Under the **Oracle-Obj Setting**, CLIPSeg performs better than CATSeg on both source and target datasets. But CATSeg is less biased to the part as it performs the best on the target dataset under the **Pred-Obj Setting**. Hence CATSeg is less prone to overfitting than CLIPSeg with even more learnable parameters. We also explore the cross-dataset part segmentation from Pascal-Part-116 to ADE20K-Part-234 for analyzing the potential failures cases in either the part boundaries delineation or the language misunderstanding. The qualitative results are left to Section C of the supplementary material.

## 6 Conclusion

In conclusion, open-vocabulary part segmentation has unique challenges: the limited availability of labeled data, the complexity of part structures and the open granularity challenge. To inspire future research, we proposed the **OV-PARTS** benchmark with newly cleaned Pascal-Part-116 and ADE20K-Part-234 datasets. And we introduced three benchmark tasks: Generalized Zero-Shot Segmentation, Cross-Dataset Segmentation, and Few-Shot Segmentation, which assesses the analogical reasoning, open granularity, and few-shot adapting abilities of OV-PARTS models. Furthermore, we improved two paradigms from existing object-level OVSS methods as the baselines. Through comprehensive experimental analysis, we provided insights into the strengths and limitations of current approaches, highlighting the potential of leveraging large foundation models for OV-PARTS.

**Acknowledgements.** This work is supported by Shanghai Artificial Intelligence Laboratory, HKU Startup Fund, HKU Seed Fund for Basic Research, and HKU Seed Fund for Translational and Applied Research. Shu Kong is supported by SRG2023-00044-FST.

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

Table A.1: Model complexity analysis on Pascal-Part-116. ZSseg+ 1/2st is first/second stage.

| Model | Backbone | Image Size | Trainable Params (M) | FLOPs (G) |
|-------|----------|------------|----------------------|-----------|
| ZSseg+ 1st | ResNet-101c | $512 \times 704$ | 61.1 | 103.9 |
| ZSseg+ 2st | ViT-B/16 | $384 \times 384$ | 0.004 | 58.9 |
| CATSeg | ResNet-101 & ViT-B/16 | $384 \times 384$ | 30.9 | 139.0 |
| CLIPSeg | ViT-B/16 | $352 \times 352$ | 1.5 | 97.9 |

# A    Implementation Details

**Two-Stage Baselines**. Except for the Object Mask Prompt and Compositional Prompt Tuning designs, we follow the default architecture in the original ZSseg paper. The number of part queries is set to 50. All the two-stage baselines are trained with AdamW optimizer with the initial learning rate of 1e-4 and weight decay of 1e-4. A poly learning rate policy with a power of 0.9 is adopted. The total batch size is 16 and the total training iteration is 20k. For the training of Compositional Prompt Tuning, a SGD optimizer with the initial learning rate of 2e-2 and weight decay of 5e-4 is used. And we adopt a warm-up cosine learning rate policy with 100 warm-up iterations. The total batch size is 32 and the total training iteration is 3k. We sample 128 training samples for each object part class. The length of the learnable object and part prompt tokens are 4. The object tokens are initialized from the text embedding of the template "a photo of". The initial value of the learnable fusion weight is 0.5.

**One-Stage Baselines**. We adopt the original architecture of both CATSeg and CLIPSeg as described in their respective papers. For finetuning CATSeg, we utilize their pretrained model with a ResNet-101 backbone. However, while CATSeg achieves the best performance by finetuning the attention layers of CLIP's visual encoder in open vocabulary object segmentation, we observe poor performance with the same finetuning strategy in OV-PARTS. In our experiments, we only finetune the backbone with a backbone multiplier of 0.1 and the swin transformer based decoder. We employ an AdamW optimizer with an initial learning rate of 2e-4, weight decay of 1e-4, and a cosine learning rate policy. The total batch size is 8, and the training iterations amount to 40k. For CLIPSeg, we utilize the same optimizer settings and learning rate policy as CATSeg. The training iterations are set to 20k for the zero-shot/cross-dataset part segmentation setting and 3k for the few-shot part segmentation setting.

**Model Complexity**. We analyze the computational complexity of these two types of baselines and summarize the number of trainable parameters and FLOPs in Table A.1. The complexity is evaluated on Pascal-Part-116. It's evident that the one-stage CLIPSeg, which solely refines a lightweight transformer decoder and employs parameter-efficient modules, showcases the fewest trainable parameters and the lowest FLOPs. In contrast, the two-stage ZSseg+ approach, involving the training of a complete maskformer with a larger resolution, leads to the highest count of trainable parameters and FLOPs.

# B    Benchmark Datasets Details

## B.1    Pascal-Part-116

Pascal-Part-116 contains 8431 training images and 850 testing images. Compared to the original version of Pascal-Part, we discard the directional indicator for some part classes and merge them to avoid overly intricate part definitions. The category vocabulary and merging details are as follows:

`aeroplane [body, stern, left/right wing, tail, engine, wheel]`

`bicycle [front/back wheel, saddle, handlebar, chainwheel, headlight]`

`bird [left/right wing, tail, head, left/right eye, beak, torso, neck, left/right leg, left/right foot]`

`bottle [body, cap]`

`bus [wheel, headlight, front, left/right side, back, roof, left/right mirror, front/back license plate, door, window]`

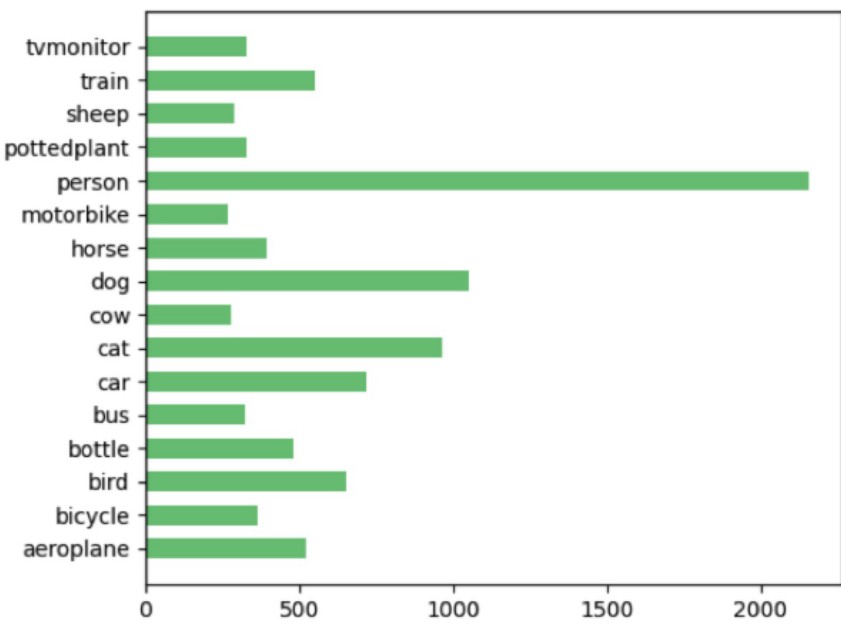

Figure A.1: The number of object masks that have corresponding part masks in Pascal-Part-116.

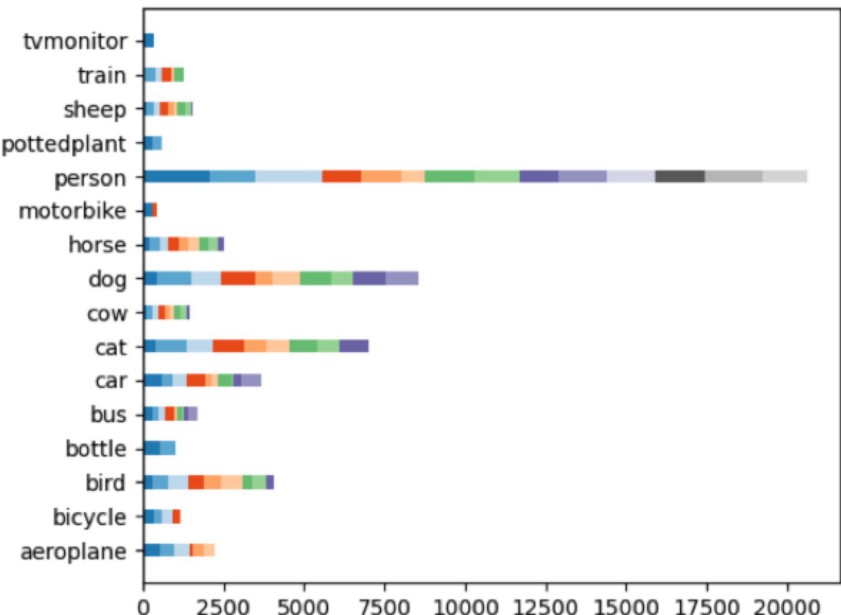

Figure B.2: The number of part masks for each object class in Pascal-Part-116. Each horizontal bar is color-coded to represent a specific part class belonging to the object. The colors of the bars are ordered from left to right based on the part sequence in the list of objects with parts.

```
car [wheel, headlight, front, left/right side, back, roof, left/right
mirror, front/back license plate, door, window]

cat [tail, head, left/right eye, torso, neck, left-front/right-front
/left-back/right-back leg, nose, left-front/right-front/left-back/right-back
paw, left/right ear]

cow [tail, head, left/right eye, torso, neck, left-front-upper/left-front-lower
/right-front-upper/right-front-lower/left-back-upper/left-back-lower/right-back-upper
/right-back-lower leg, left/right ear, muzzle, left/right horn]

dog [tail, head, left/right eye, torso, neck, left-front/right-front
/left-back/right-back leg, nose, left-front/right-front/left-back/right-back
paw, left/right ear, muzzle]

horse [tail, head, left/right eye, torso, neck, left-front-upper/left-front-lower
/right-front-upper/right-front-lower/left-back-upper/left-back-lower/right-back-upper
/right-back-lower leg, left/right ear, muzzle, left-front/right-front/left-back
/right-back hoof]

motorbike [front/back wheel, saddle, handlebar, headlight]

person [head, left/right eye, torso, neck, left-lower/right-lower/left-upper/right-upper
leg, foot, nose, left/right ear, left/right eyebrow, mouth, hair,
left/right lower arm, left/right upper arm, left/right hand]

pottedplant [pot, plant]

sheep [tail, head, left/right eye, torso, neck, left-front-upper/left-front-lower
/right-front-upper/right-front-lower/left-back-upper/left-back-lower/right-back-upper
/right-back-lower leg, left/right ear, muzzle, left/right horn]

train [headlight, head, front, left/right side, back, roof, coach]

tvmonitor [screen]
```

The unseen objects are colored blue and the removed terms are colored purple.

## B.2 ADE20K-Part-234

The original subset of SceneParse150 comprises 20,210 training images and 2,000 validation images. After filtering out less frequent parts, the subset is reduced to 7,347 training images and 1,016 validation images. In ADE20K, most object parts have sparse mask annotations, and only a subset of object instances have part annotations. Hence, ADE20K-Part-234 provides the instance-level object mask annotations along with their part mask annotations. To maximize the use of labeled data and ensure authentic evaluations, different data splits are designed for the three task settings. (1) Generalized Zero-Shot Part Segmentation: Models are trained on the seen object instances from the 7,347 training images. Testing is performed on both unseen object instances from the same 7,347 training images and all object instances from the 1,016 validation images. (2) Few-Shot Part Segmentation: For each object class, 16 training images are sampled following the approach in Pascal-Part-116. we adapt the validation set from the generalized zero-shot part segmentation setting by removing the images that occur in the sampled 16-shot training set. (3) Cross-Dataset Part Segmentation: The original data split (7347/1016 training/validation images) is used since we mainly test on the Pascal-Part-116 dataset. The annotated objects with their parts are listed as follows:

```
person [arm, back, foot, gaze, hand, head, leg, neck, torso]
door [door frame, handle, knob, panel]
clock [face, frame]
toilet [bowl, cistern, lid]
cabinet [door, drawer, front, shelf, side, skirt, top]
sink [bowl, faucet, pedestal, tap, top]
lamp [arm, base, canopy, column, cord, highlight,light source, shade,tube]
sconce [arm, backplate, highlight, light source, shade]
chair [apron, arm, back, base, leg, seat, seat cushion, skirt, stretcher]
chest of drawers [apron, door, drawer, front, leg]
```

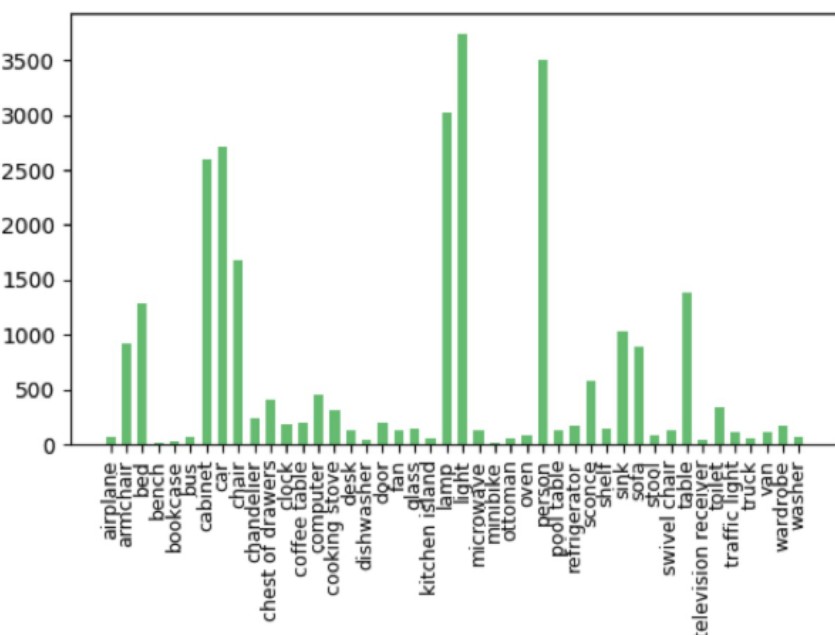

Figure B.3: The statistics for the number of object masks with part masks on ADE20K-Part-234.

chandelier [arm, bulb, canopy, chain, cord, highlight, light source, shade]
bed [footboard, headboard, leg, side rail]
table [apron, drawer, leg, shelf, top, wheel]
armchair [apron, arm, back, back pillow, leg, seat, seat base,seat cushion]
ottoman [back, leg, seat]
shelf [door, drawer, front, shelf]
swivel chair [back, base, seat, wheel]
fan [blade, canopy, tube]
coffee table [leg, top]
stool [leg, seat]
sofa [arm, back, back pillow, leg, seat base, seat cushion, skirt]
computer [computer case, keyboard, monitor, mouse]
desk [apron, door, drawer, leg, shelf, top]
wardrobe [door, drawer, front, leg, mirror, top]
car [bumper, door, headlight, hood, license plate, logo, mirror, wheel,
window, wiper]
bus [bumper, door, headlight, license plate, logo, mirror, wheel, window,
wiper]
oven [button panel, door, drawer, top]
cooking stove [burner, button panel, door, drawer, oven, stove]
microwave [button panel, door, front, side, top, window]
refrigerator [button panel, door, drawer, side]
kitchen island [door, drawer, front, side, top]
dishwasher [button panel, handle, skirt]
bookcase [door, drawer, front, side]
television receiver [base, buttons, frame, keys, screen, speaker]
glass [base, bowl, opening, stem]
pool table [bed, leg, pocket]
van [bumper, door, headlight, license plate, logo, mirror, taillight, wheel,
window, wiper]
airplane [door, fuselage, landing gear, propeller, stabilizer, turbine
engine, wing]
truck [bumper, door, headlight, license plate, logo, mirror, wheel,
windshield]
minibike [license plate, mirror, seat, wheel]

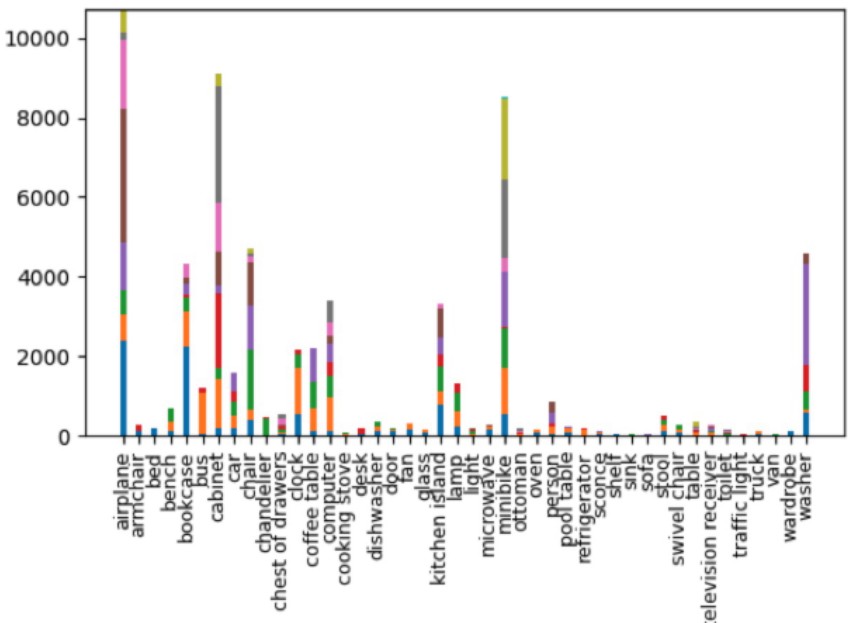

Figure B.4: The number of part masks for each object class in ADE20K-Part-234. Each horizontal bar is color-coded to represent a specific part class belonging to the object. The colors of each bar are ordered from bottom to top according to the part sequence in the list of objects with parts.

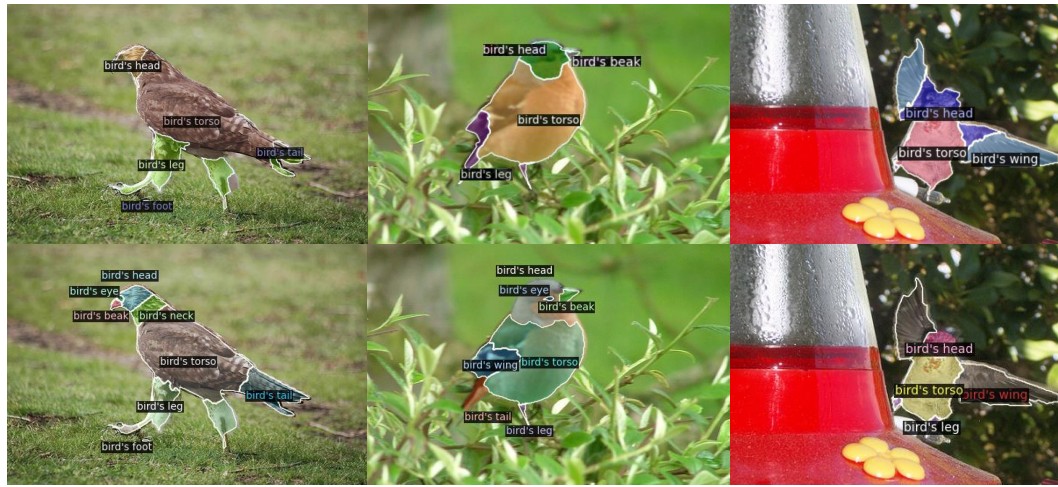

Figure B.5: Qualitaive results on **ZSseg+**, **CATSeg** and **CLIPSeg** concerning the challenging unseen "bird" class in Pascal-Part-116, as shown in the first row. The second row shows the corresponding ground truth. We can observe that CATSeg and CLIPSeg can generalize to the more novel parts: "Bird's Beak" and "Bird's Wing"

```
washer [button panel, door, front, side]
bench [arm, back, leg, seat]
traffic light [housing, pole]
light [aperture, canopy, diffusor, highlight, light source, shade]
```

## B.3 Data Statistics Analysis.

We report the statistics for the number of object masks that have part annotations in Pascal-Part-116 (see Figure A.1) and ADE20K-Part-234 (see Figure B.3). The total number of part masks for each object and the proportion of each part are shown in Figure B.2 (Pascal-Part-116) and Figure B.4 (ADE20K-Part-234). In Figure B.2, the color sequence from left to right corresponds to the part word sequence as listed in Section B.1. In Figure B.4, the color sequence from bottom to up corresponds to

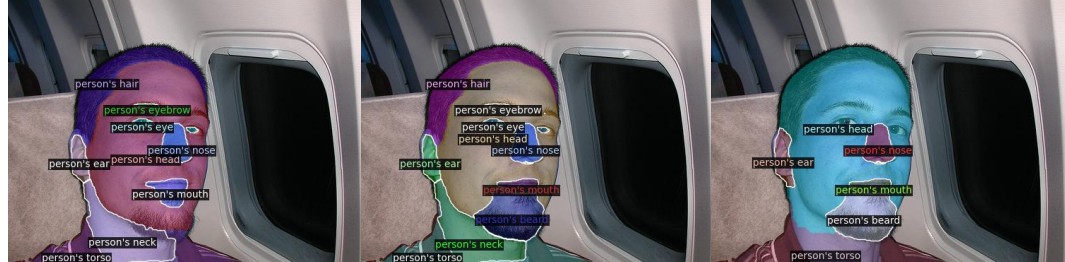

Figure B.6: Qualitative results on CATSeg's multi-granular generalization ability. From the left to the middle image, the model generalizes from "head" to the more fine-grained "beard". From the middle to the right image, the model generalizes from ["hair", "eyebrow", "eye"] to the coarse-grained "head" and also from "neck" to "torso".

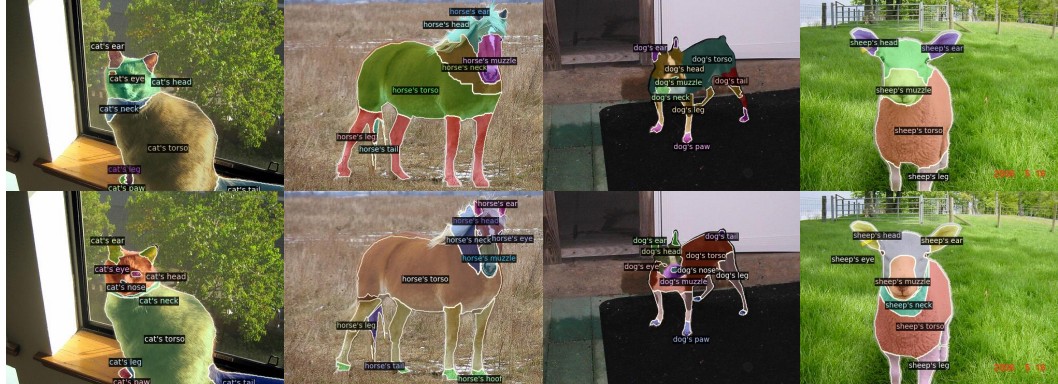

Figure B.7: More qualitative results of generalized zero-shot part segmentation on Pascal-Part-116 are in the first row. The ground truth is in the second row. The seen classes are "cat" and "horse" while the unseen classes are "dog" and "sheep".

the part word sequence as listed in Section B.2. Additionally, we report the scale distribution for the part masks of each object as shown in Figure B.8.

## C  Qualitative Results

The qualitative results on the comparison among **ZSseg+**, **CATSeg** and **CLIPSeg** for the challenging case "bird" are shown in Figure B.5. Figure B.6 shows the multi-granular generalization ability of the one-stage baselines. The adopted model is CATSeg. The visualization sample is from the "person" class in Pascal-Part-116. We give more qualitative results on Pascal-Part-116 and ADE20K-Part-234 on the three proposed task settings. The adopted model is CLIPSeg with finetuning (VA+L+F+D). The visualization results for the **Generalized Zero-Shot Part Segmentation** on Pascal-Part-116 and

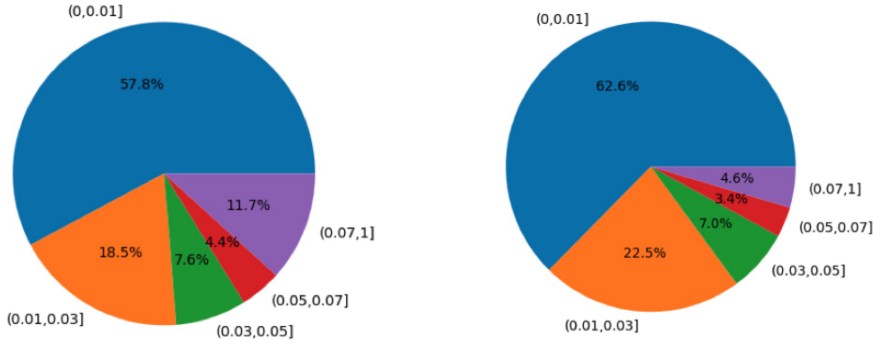

Figure B.8: The scale ratio (number of pixels in the object part mask out of all pixels in an image.) distribution of all part masks of Pascal-Part-116 (Left) and ADE20K-Part-234 (Right).

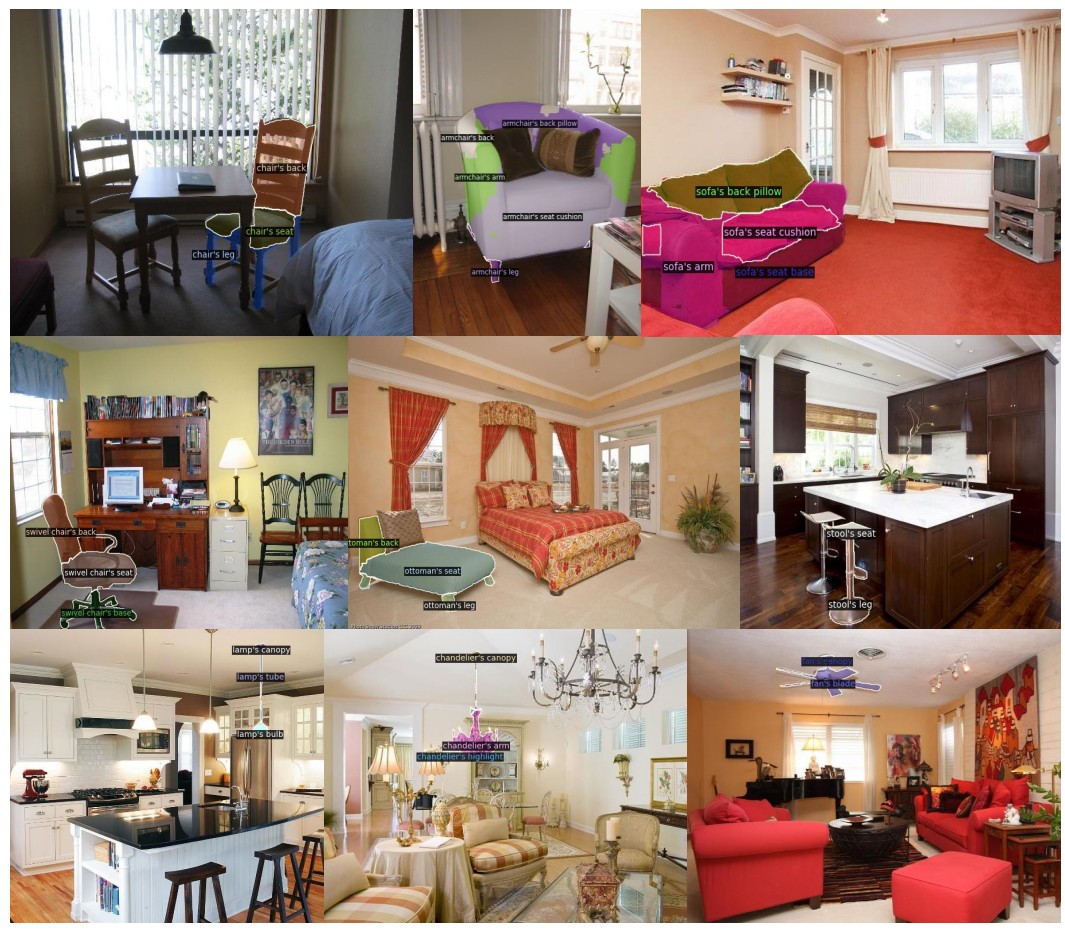

Figure B.9: Qualitative results of generalized zero-shot part segmentation on ADE20K-Part-234. The first and second rows show the generalize from the seen classes [chair, armchair, sofa] to the unseen classes [swivel chair, ottoman, stool]. The third row shows the generalize from the seen classes [lamp, chandelier] to the unseen class [fan]. Notably, "fan's blade" is novel at the object and part level.

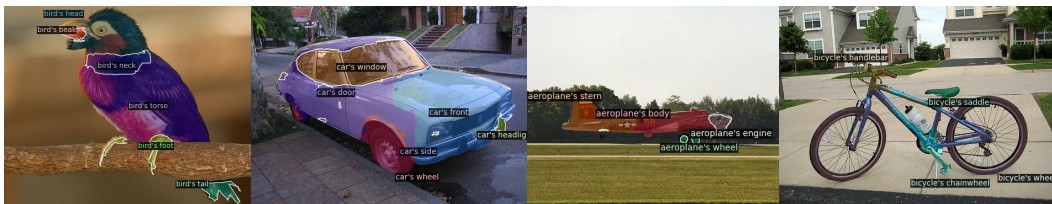

Figure B.10: Qualitative results of few-shot part segmentation on Pascal-Part-116. We display the segmentation map of four classes: "bird", "aeroplane", "car" and "bicycle".

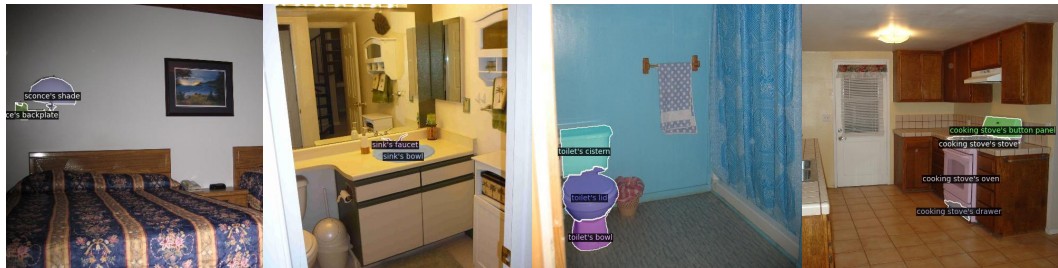

Figure B.11: Qualitative results of few-shot part segmentation on ADE20K-Part-234. We display the segmentation map of four classes: "lamp", "sink", "toilet" and "cooking stove".

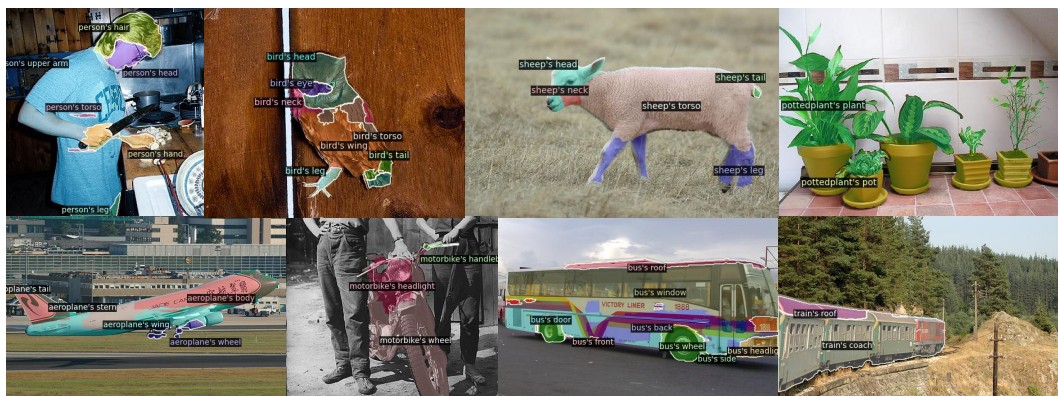

Figure B.12: Qualitative results of cross-dataset part segmentation on Pascal-Part-116. Pascal-Part-116 provides more fine-grained part annotations for the "person" category, such as "hair" and "upper arm". The model trained on ADE20K-Part-234 demonstrates the ability to recognize "hair" but struggles to generalize from "arm" to "upper arm" and "lower arm" accurately. Moreover, the model exhibits potential in generalizing parts of the "airplane" category. Although ADE20K-Part-234 annotates the parts as "door", "fuselage", "landing gear", "propeller", "stabilizer", "turbine engine", and "wing", the model can generalize them to Pascal-Part-116's parts, including "body", "stern", "wing", "tail", "engine", and "wheel", despite the differences in vocabulary and granularity. Notably, ADE20K-Part-234 does not contain related classes to "bird", "sheep", and "potted plant", but the model demonstrates a certain level of generalization ability to segmenting these categories.

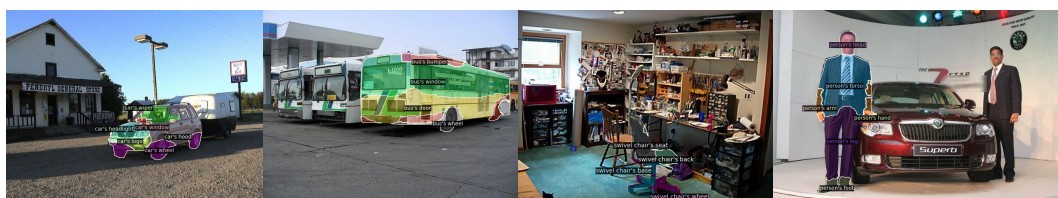

Figure B.13: Qualitative results of cross-dataset part segmentation on ADE20K-Part-234. For the categories "car" and "bus", the part annotations in Pascal-Part-116 are more coarse-grained. When tested on ADE20K-Part-234, the model trained on Pascal-Part-116 can predict novel parts like "logo", "wiper", "hood", and "bumper". However, the segments and part labels don't align accurately. For example, the model still segments the "bus's roof", which is annotated in Pascal-Part-116, but wrongly assigns it to "bus's bumper" in ADE20K-Part-234. This showcases the challenge of generalizing across different granular part definitions. For the novel object "swivel chair", the model adeptly delineates part boundaries even without relevant objects in Pascal-Part-116. But the category errors are still present. In the case of the "person" category, the model only segments the "upper arm", which demonstrates the difficulty of generalizing from "upper/lower arm" to "arm".

ADE20K-Part-234 are shown in Figure B.7 and Figure B.9 respectively. We report the qualitative results for the **Few-Shot Part Segmentation** on Pascal-Part-116 in Figure B.10 and on ADE20K-Part-234 in Figure B.11. And the results for the **Cross-Dataset Part Segmentation** on Pascal-Part-116 are shown in Figure B.12. Furthermore, we present the qualitative results for models trained on Pascal-Part-116 and then tested on ADE20K-Part-234 are shown in Figure B.13.

## D  Future Works and Negative Societal Impacts

Although part-level OVSS indeed presents more challenges compared to object-level OVSS, the OV-PARTS benchmark datasets have lower quality than existing object-level OVSS benchmark datasets. The original version of Pascal-Part and ADE20K-Part are annotated without considering the open vocabulary scenario especially the analogical reasoning ability and open granularity ability that we care about in a part-level OVSS model. The benchmark datasets need to be continuously expanded and improved to encompass more diverse and complex object-part annotations. There may be potential negative societal impacts associated with the OV-PART benchmark. The deployment of fine-grained part segmentation models in various real-world applications may lead to unintended consequences. We must ensure that the predictions be reliable and accurate in critical applications, such as medical diagnosis or autonomous vehicles. Also, there is a possibility of misuse of part segmentation technology for malicious purposes, such as creating deepfake images or spreading misinformation. Ensuring security measures and appropriate regulations to prevent such misuse is vital in the development and deployment process.

