# OV-PARTS: Towards Open-Vocabulary Part Segmentation
## *(Supplementary Material)*

**Coauthor**

The supplementary material is organized as follows:

- Implementation Details.(Sec. A)
- Details of Benchmark Datasets: Pascal-Part-116 and ADE20K-Part-234 (Sec. B).
- Qualitative Results of Three Benchmark Tasks (Sec. C).
- Future Works and Negative societal Impacts (Sec. D).