# OpenReview forum: "OV-PARTS: Towards Open-Vocabulary Part Segmentation"
_NeurIPS.cc/2023/Track/Datasets_and_Benchmarks — NeurIPS 2023 Datasets and Benchmarks Poster_

### Official Review · Reviewer_v7gi · 2023-07-21

**Rating:** 6
**Confidence:** 3
**Clarity:** Good.

**Strengths:**

1. The author has conducted a large number of comparative experiments on the proposed task, which will help researchers to understand the gap between various methods.
2. The author proposes a benchmark for part segmentation and conducts systematic experiments, task design, and dataset construction, which will contribute to the development of the community.
3. The writing and readability of the article are good.

**Additional Feedback:**

1. The author only briefly explained the construction of the dataset in Section 3.1, and the difference between the constructed dataset and the original dataset is not clear.
2. The author constructed a new dataset, but lacked statistical comparison with similar datasets.
3. There are some typos in the paper, such as line 234 'Part-Part-116', it is recommended to check the full text.
4. The '-' in the table lacks a description.

**Correctness:**

The author only has a brief description of the specific construction method of the dataset.

**Documentation:**

There is no corresponding connection to the dataset in the paper.

**Ethics:**

Yes.

**Limitations:**

Not applicable.

**Opportunities For Improvement:**

See other feedback.

**Relation To Prior Work:**

It is reasonable to have a corresponding discussion.

**Summary And Contributions:**

The author proposes a benchmark for the open vocabulary part segmentation task, which mainly contains two new datasets: Pascal-Part-116 and ADE20K-Part-234,  and three specific tasks: Generalized Zero-Shot Part Segmentation, Cross-Dataset Part Segmentation, and Few-Shot Part Segmentation. And carried out a detailed experiment.

---

> ### Author Response · Authors · 2023-08-18
>
> ### Q1. Construction details of the datasets and the difference with the original datasets.
>
> For Pascal-Part-116,  we merge the parts by discarding directional terms like "left," "right," "front," "back," "upper," and "lower". The details are shown in the updated supplement material Section B.1.
>
> For ADE20K-Part-234, we first filter the annotations with scripts as said in L129-L131, and merge some parts to mitigate issues related to data sparsity and noise. The mapping from object and part classes in our ADE20K-Part-234 to original ids in ADE20K is provided in the following format (A merged object or part will correspond to multiple class ids in ADE20K):
>
> ```json
> {
> "door": { # object class name
>           "id": [783, 774], # object class id in ADE20K
>           "parts": { # part class name and its class id in ADE20K
>               "panel": [1746, 1748, 774],
>               "door frame": 776,
>               "handle": 1180,
>               "knob": 1380
>           },
> ...
> }
> ```
> The complete JSON file is provided in our code release here: https://github.com/kellyiss/OV_PARTS/blob/main/baselines/data/datasets/ade20kpart234_mapping.json .
>
> ### Q2. Comparison with similar datasets.
>
> There are only four datasets that annotate both objects and parts: Pascal-Part, ADE20K-Part, PartImageNet[1], and PACO-LVIS[2].
> For PartImageNet, although it consists of 158 classes from ImageNet, the part diversity is much lower compared to the objects. And PartImageNet doesn't provide object masks.
>
> | Category | Annotated Parts |
> |--------------|-----------------------|
> | Quadruped (46) | Head, Body, Foot, Tail |
> | Biped (17) | Head, Body, Hand, Foot, Tail |
> | Fish (10) | Head, Body, Fin, Tail |
> | Bird (14) | Head, Body, Wing, Foot, Tail |
> | Snake (15) | Head, Body |
> | Reptile (20) | Head, Body, Foot, Tail |
> | Car (23) | Body, Tier, Side Mirror |
> | Bicycle (6) | Head, Body, Seat, Tier |
> | Boat (4) | Body, Sail |
> | Aeroplane (2) | Head, Body, Wing, Engine, Tail |
> | Bottle (5) | Body, Mouth |
>
> PACO-LVIS was not adopted for OV-PARTS because it primarily annotates small objects like handheld devices, accessories, containers, and tools which often appear tiny in the natural scene images.
> According to their paper, 80% of object-part masks are small.
>
> Even the supervised model with a ResNet-101 backbone can achieve only 13.7 mask AP for part segmentation.
> The significant part segmentation bottleneck makes PACO-LVIS less applicable to the OV-PARTS.
>
> ### Q3. The '-' in the table lacks a description.
>
> For the supervised reference MaskFormer, the harmonic-IoU metric, which is designed for the zero-shot setting, is intentionally ignored.
>
> For the two-stage baselines in Table 1 and Table 2, the omission of ZSseg+/ResNet-50/CPTCoCoOp under Pred-Obj is intentional, as it was designed as an ablation study on CPTCoOp and CPTCoCoOP. To simplify the analysis, the results are only analyzed under Oracle-Obj, as both models use the same object model.
>
> Furthermore, in Table 2, we apologize for the missing result for ZSseg+/ResNet-101c/CPTCoOp and have added it to the revised version. For the one-stage baselines, their design enables them to generalize directly to both object-level and part-level without the need for additional object models. The reported results for CATSeg and CLIPSeg without finetuning under the Oracle-Obj setting serve the purpose of showing their impressive zero-shot part performance. In fact, it's not a fair comparison with other baselines that are finetuned on the part datasets. Hence, we choose to skip the results under the Pred-Obj setting.

---

### Official Review · Reviewer_VdPY · 2023-07-25
**Review for "OV-PARTS: Towards Open-Vocabulary Part Segmentation"**

**Rating:** 6
**Confidence:** 5
**Correctness:** Yes.
**Clarity:** This paper is well written.

**Strengths:**

- As shown in Fig. 2, granularity gaps between the methods are important problems, but have been yet handled.
- The paper categories challenging open-vocabulary settings for part segmentation, including generalized zero-shot part segmentation, cross-dataset part segmentation, and few-shot part segmentation, which makes sense.
- Not only the paper evaluates various methods, but also it analyzes the ablation of the previous method, as shown in Tab. 3 or 4.
- The paper is well written.


**Additional Feedback:**

Please refer "*Opportunities For Improvement" section.

**Documentation:**

Yes.

**Ethics:**

No.

**Limitations:**

Please refer the above section.

**Opportunities For Improvement:**

- The paper proposed Pascal-Part-116 and ADE20K-Part-234 based on their own benchmarks. For open-vocabulary part segmentation, the authors manually merged some class, or revised some labels. But detailed explanation is required. For instance, how did the authors choose the merged class labels? Moreover, why did the authors choose PASCAL-Part and ADE20K, even though there exist more than 6 benchmark for evaluating open-vocabulary segmentation. It would be better if more detailed explanation about dataset is provided.
- In Tab. 1, MaskFormer seems state-of-the-art, but it is not proposed to solve the open-vocabulary setting. Could the authors explain why?
- In Tab. 1 and 2, ZSseg, CATSeg, and CLIPSeg used different backbone architecture, so it's really hard to compare in a fair setting. It would be better if the same backbone is considered.
- Computation complexity analysis is required.

**Relation To Prior Work:**

It clearly discussed how this work differs from previous contributions.

**Summary And Contributions:**

This paper proposed an open-vocabulary part segmentation (OV-PARTS) benchmark. First of all, the authors used two new versions of publicly available datasets such as Pascal-Part-116 and ADE20K-Part-234. Furthermore, three specific tasks are considered, such as generalized zero-shot part segmentation, cross-dataset part segmentation, and few-shot part segmentation. The paper intensively evaluated two categories for open-vocabulary segmentation tasks, including two-stage baseline and one-stage baseline.

---

> ### Author Response · Authors · 2023-08-18
>
> ### Q1. Details about the dataset reorganization.
>
> For Pascal-Part-116,  we merge the parts by discarding directional terms like "left," "right," "front," "back," "upper," and "lower". The details are shown in the updated supplement material Section B.1.
>
> For ADE20K-Part-234, we first filter the annotations with scripts as said in  L129-L131, and merge some parts to mitigate issues related to data sparsity and noise. The mapping from object and part classes in our ADE20K-Part-234 to original ids in ADE20K is provided in the following format (A merged object or part will correspond to multiple class ids in ADE20K):
>
> ```json
> {
> "door": { # object class name
>           "id": [783, 774], # object class id in ADE20K
>           "parts": { # part class name and its class id in ADE20K
>               "panel": [1746, 1748, 774],
>               "door frame": 776,
>               "handle": 1180,
>               "knob": 1380
>           },
> ...
> }
> ```
> The complete JSON file is provided in our code release here: https://github.com/kellyiss/OV_PARTS/blob/main/baselines/data/datasets/ade20kpart234_mapping.json .
>
> ### Q2. Why choose PASCAL-Part and ADE20K.
>
> There are only four datasets that annotate both objects and parts: Pascal-Part, ADE20K-Part, PartImageNet[1], and PACO-LVIS[2].
> For PartImageNet, although it consists of 158 classes from ImageNet, the part diversity is much lower compared to the objects. And PartImageNet doesn't provide object masks.
>
> | Category | Annotated Parts |
> |--------------|-----------------------|
> | Quadruped (46) | Head, Body, Foot, Tail |
> | Biped (17) | Head, Body, Hand, Foot, Tail |
> | Fish (10) | Head, Body, Fin, Tail |
> | Bird (14) | Head, Body, Wing, Foot, Tail |
> | Snake (15) | Head, Body |
> | Reptile (20) | Head, Body, Foot, Tail |
> | Car (23) | Body, Tier, Side Mirror |
> | Bicycle (6) | Head, Body, Seat, Tier |
> | Boat (4) | Body, Sail |
> | Aeroplane (2) | Head, Body, Wing, Engine, Tail |
> | Bottle (5) | Body, Mouth |
>
> PACO-LVIS was not adopted for OV-PARTS because it primarily annotates small objects like handheld devices, accessories, containers, and tools which often appear tiny in the natural scene images.
> According to their paper, 80% of object-part masks are small.
>
> Even the supervised model with a ResNet-101 backbone can achieve only 13.7 mask AP for part segmentation.
> The significant part segmentation bottleneck makes PACO-LVIS less applicable to the OV-PARTS.
>
> ### Q3. MaskFormer baseline in Table 1.
>
> The MaskFormer in Table 1 and Table 2 is a supervised baseline (trained on data of both seen and unseen classes), provided as a reference for zero-shot performance evaluation.
> The two-stage baseline ZSseg is a method that combines MaskFormer and CLIP for open vocabulary segmentation.
>
> ### Q4. Different backbone architecture.
>
> CLIPSeg adopts a frozen CLIP visual encoder and trains the decoder on the PhraseCut dataset.
> CATSeg adopts both a frozen CLIP visual encoder and a trainable ResNet backbone, training the decoder on COCO.
> Zsseg+ trains the MaskFormer on the part datasets from scratch in the first stage.
> Hence, it's indeed hard to achieve absolute fairness with foundation models due to the variations in their architectures and training dataset. We basically follow the existing OVSS works to specify the feature backbone.
>
> ### Q5. Computation complexity.
>
> We have added the model complexity analysis in supplement material Table A.1.
>
> [1] He, Ju et al. “PartImageNet: A Large, High-Quality Dataset of Parts.” ArXiv abs/2112.00933 (2021).
>
> [2] Ramanathan, Vignesh et al. “PACO: Parts and Attributes of Common Objects.” ArXiv abs/2301.01795 (2023).

---

### Official Review · Reviewer_cNC2 · 2023-07-26
**New benchmark and baselines for open-vocabulary part segmentation**

**Rating:** 7
**Confidence:** 3

**Strengths:**

- Object part datasets are rare, so adding a clean benchmark will be valuable to the community.
- Baselines are strong, there are clear improvements to the existing methods on the new benchmarks.
- Many things are tested and evaluated in different ways so the information density in the paper is high.

**Additional Feedback:**

No

**Clarity:**

Some open questions remain:

- Does the union of all parts always cover the entire object for the two datasets? Or is there sometimes unspecified parts left, or some pixels annotated as more than one part? E.g. supplementary line 53, "tvmonitor" only has "screen" as a part - what happens to the plastic border of the television, how is it annotated?

- Exact definition of "over-segmentation parts"

- What is the "Aggregate" operation on line 197 do mathematically?

**Correctness:**

There is no code submitted so there is no way of checking if any of the claims are correct. Please either provide the code to reproduce the experiments, or provide a reasonable explanation of why you will not provide the code.

**Documentation:**

Yes

**Limitations:**

see Opportunities For Improvement

**Opportunities For Improvement:**

No new annotations are created, the authors only filter existing data and merge class labels, so the novelty in the two datasets is limited. Maybe some more details on these changes could help: Some visual examples of the failure cases in the original dataset and their solution in the new dataset. For example, line 117 claims "unnecessary to discern between the semantics of labels such as “cow’s left front lower leg” and “cow’s left front upper leg" - why is this unnecessary? What is the goal of the new subsets and why do these semantics not fit the goal?

I found the definitions of the three benchmark types to be very confusing due to several reasons I will outline below.

- In most other works, zero-shot means not training on the train set at all, but instead training on other data (e.g. training CLIP on 400M noisy web images and then applying it "zero-shot" to imagenet dataset.) However here you define a zero-shot setting with a training set. Supp line 29: you remove the annotation for the unseen objects, supp line 62: You train only on the seen object instances. In my opinion if you train on the training set this setting can be called open-vocabulary but not zero-shot.

- Line 231: You choose different metrics for the 3 benchmark modes. Maybe instead of harmonic mean you could report IoU over all objects so the "generalized zero-shot" setting is comparable to the other 2.

- Table 5 is confusing: Is the zero-shot setting reported on oracle or pred objects? Most importantly why does the few-shot setting perform worse than the zero-shot setting? Do you only sample 16 parts for each possible object-part combination, or do you take the existing zero-shot training set and add more parts to get at least 16 for each?

- Why do you add "One shot" in table 3? To me it does not make sense to change the setting for one of the experiments. You introduce basically a fourth setting "Generalized zero-shot but with one-shot" which makes everything even more complicated. In my opinion you should leave this out.

- Maybe you can merge the descriptions of the benchmarks in 3.2 and 5.1 and for each describe exactly what the training and test sets are. I found myself having to jump between those two and the supplementary alot trying to figure out how your data actually looks like. Benchmarks should be easily runnable in a "fair" manner (knowing exactly what one is allowed to do and what not). Currently this missing clarity is the biggest weakness of the paper in my opinion.

- Your paper does suffers a bit from doing too many things at once. An example of this is figure 1, where you try to show all 3 benchmark settings as well as qualitative results in one figure, leading to the figure being crowded with too small text.

**Relation To Prior Work:**

Yes

**Summary And Contributions:**

Authors propose a new benchmark and baselines to test object part recognition:

- Merging some of the part classes of Pascal-Part to create a more practical version Pascal-Part-116
- Filter objects and merge some part classes of SceneParse150 to create a cleaner subset named ADE20K-Part-234
- Define three benchmark settings "Generalized zero-shot part segmentation", "Cross-dataset part segmentation" and "few-shot part segmentation."
- Apply and modify two existing methods as baselines.

---

> ### Author Response · Authors · 2023-08-18
>
> ### Q1. Definition of over-segmentation parts such as  "cow’s left front lower leg" and "cow’s left front upper leg".
>
> Pascal-Part annotations use directional terms like "left," "right," "front," "back," "upper," and "lower" to label parts as shown in the updated supplement material Section B.1.
> It can be challenging to accurately segment the intricate part boundaries and distinguish between the directions.
> The closed-set part segmentation methods address this challenge by following the merging rules of [1] for animals, [2] for vehicles, and [3] for human bodies which results in the commonly used Pascal-Part-58.
>
> Our Pascal-Part-116 has offered a higher level of granularity compared to the closed-set setting.
> But these directional indicators are not necessary for OV-PARTS.
> They not only create a bottleneck in part segmentation but also cause overfitting with sparse data, which hinders effective language-driven generalization.
> As shown in the updated supplement material Figure B.12 1st image, the model trained on "person's upper/lower arm" can't even generalize to "person's arm" accurately, which indicates a significant issue of overfitting.
> To address the complexities of part-level OVSS, we think such modifications are more meaningful and practical.
>
> ### Q2. Confusion about generalized zero-shot setting.
>
> Traditional zero-shot learning aims to predict the samples from unseen categories, while generalized zero-shot learning aims to predict the samples from both seen classes and unseen classes.
> For models trained on massive data like CLIP,  since there is no task gap when applying to the target classification dataset like ImageNet, no further training is required.
> But most open vocabulary semantic segmentation models require training on limited segmentation data in order to transfer the power of CLIP to pixel-level prediction.
>
> Generalized zero-shot segmentation and cross-dataset segmentation are two popular ways to validate a segmentation model's open vocabulary ability. For example, ZegFormer[4] is evaluated only under the generalized zero-shot setting (split the classes into seen/unseen set) while OpenSeg[5] is evaluated only under the cross-dataset setting (trained on COCO and tested on Pascal VOC and ADE20K). ZSseg[6] is evaluated under both settings.
>
> ### Q3. Choose harmonic mean for the generalized zero-shot setting.
>
> In the generalized zero-shot setting, the objective is to achieve balanced performance between both seen and unseen classes.
> Using mean-IoU as the evaluation metric could lead to the dominance of high seen class accuracy, thus overshadowing the performance on the typically lower unseen class accuracy.
> It's noteworthy that the mean-IoU of two-stage baselines is higher than that of one-stage baselines, because of training a state-of-the-art maskformer on seen classes. However, it's evident that one-stage baselines are better aligned with the open vocabulary setting.
>
> Indeed, the harmonic mean is a widely used metric in the generalized zero-shot setting.
> It takes into account both high and low values and penalizes extreme discrepancies between seen and unseen class accuracies caused by the unintended model overfitting in generalized zero-shot learning.
>
> ### Q4. Explanations about Table 5.
>
> a. The zero shot setting is reported on oracle object. We have added it to Table 5.
>
> b. In the few-shot setting, each object-part class is constrained by a limited number of 16 samples, aiming for few-shot prompting to the foundation models. Therefore, it's justifiable that the few-shot setting yields lower performance than the zero-shot setting.
> Because in the zero-shot setting,  the majority of classes are seen classes which have sufficient training data thus contributing to higher mIoU.
>
> [1]  Jianyu Wang and Alan L Yuille. Semantic part segmentation using compositional model combining shape and appearance. In CVPR, pages 1788–1797, 2015.
>
> [2] Yafei Song, Xiaowu Chen, Jia Li, and Qinping Zhao. Embedding 3d geometric features for rigid object part segmentation. In ICCV, pages 580–588, 2017
>
> [3] Liang-Chieh Chen, Yi Yang, Jiang Wang, Wei Xu, and Alan L Yuille. Attention to scale: Scale-aware semantic image segmentation. In CVPR, pages 3640–3649, 2016.
>
> [4] Ding, Jian, et al. "Decoupling zero-shot semantic segmentation." Proceedings of the IEEE/CVF Conference on Computer Vision and Pattern Recognition. 2022.
>
> [5] Ghiasi, Golnaz, et al. "Scaling open-vocabulary image segmentation with image-level labels." European Conference on Computer Vision 2022.
>
> [6] Xu, Mengde, et al. "A simple baseline for open-vocabulary semantic segmentation with pre-trained vision-language model." European Conference on Computer Vision 2022.

---

> ### Author Response · Authors · 2023-08-18
>
> ### Q5. Add "One shot" to Table 3.
>
> Table 3 serves as an ablation study for ZSseg+ rather than presenting the results under the zero-shot setting. The inclusion of One Shot results aims to highlight one of the advantages of ZSseg+, *i.e.*, achieving performance gains with minimal data and training costs.
>
> ### Q6. Organization of the paper.
>
> Thanks for the insightful advice to make our paper more organized. We've restructured Section 3.2 and Section 5.1.  Indeed, OV-PARTS covers too many elements, including the two new versions of datasets, three benchmark task settings, two paradigms of baselines based on several OVSS methods, and exploration of effective and efficient finetuning strategies. Also, there are many important and interesting visualizations to be displayed. To ensure clarity without overwhelming the main paper, we've included more essential details discussed by all the reviewers in our supplementary material.
>
> ### Q7. Does the union of all parts always cover the entire object or are some pixels annotated as more than one part?
>
> The part segmentation datasets are commonly featured by incomplete annotations.
> Both Pascal-Part and ADE20K annotate parts that may not cover the entire object.
> But in the open vocabulary setting, there is no "no-object" class, and the metrics are only computed on the foreground (annotated) pixels.
>
> Some OVSS methods like Lseg set an explicit class word "others" to align pixels not in the given class set. However, more methods use a learnable background vector during training, alongside class-specific CLIP text embeddings. This background vector is disregarded when calculating scores during testing.
> Each pixel is assigned only one part class. For example, a pixel of "eye" on "head" is only classified as "eye".
>
> ### Q8. "Aggregate" operation.
>
> In L208-L212, we have revised the formula to enhance its clarity.
> Firstly, we compute the class-aware segmentation masks as in the original Maskformer (biased to the seen objects). Then we binarize it and relabel each binary mask by sending it to CLIP and then fuse the two score maps to obtain the final result.

---

> > ### Comment · Reviewer_cNC2 · 2023-08-25
> >
> > Thanks for your insightful responses. I have some followup remarks:
> >
> > Q1: Your explanation for merging makes sense. However the part "person's upper arm" in figure B12 you referenced is hard to read, due to indistinguishable coloring it is not clear which pixels are segmented as arm and which as torso. In general the appendix figure quality is rather low: different font sizes, some texts are stretched vertically e.g. the toilet parts in B.11, some texts are hidden behind others e.g. aeroplane in figure B.10.
> >
> > Q5: I still think “one shot” training on unseen classes is not meaningful, since it just means that the classes are not unseen anymore. If you want to show that ZSseg+ does especially well in one-shot learning you would have to compare it with other models in the one-shot setting and show that the improvements from zero to one shot are bigger than for the other models. However this would then be a new setting: “Generalized one-shot part segmentation”. Also, you mention: “Table 3 serves as an ablation study for ZSseg+ rather than presenting the results under the zero-shot setting.” If it is not under the zero-shot setting, which setting is it? I understood that all rows are in zero-shot setting except the one-shot row, please correct me if I am wrong. In general it seems unusual to change the data or benchmark setting during an ablation on the model strategies, I would expect the data and benchmark to be fixed.
> >
> > Q6: In my review I mentioned figure 1 being crowded with too small text. To give an example for the small text, the bird parts under "pascal - unseen" are not readable even at full zoom. In your response you mention "OV-PARTS covers too many elements" and this figure also shows this, as its content would need more of a half or full page to be properly displayed. You could consider using less images for the figure (one image per dataset instead of several) or split the figure into several ones, currently you are showing 2 datasets, 3 evaluation procedures and qualitative results in the same figure.

---

> ### Author Response · Authors · 2023-08-28
>
> The visualization with text for part masks poses significant challenges when using automatic tools (e.g. detectron2 visualizer), since part masks often congregate closely and can be tiny. The workable way that we find is manual calibration of the text placement for each image. It demands considerable effort but we truly agree that qualitative results are much better now and can provide more insights.  Please check the updated supplement material. If we miss anything, we earnestly invite your observations.
>
> Q1. In Figure B.12 1st image (generalize from person's arm to person's upper/lower arm), the model gives erroneous segmentation which recognizes both "person's torso" and "person's lower arm" to "person's upper arm". We think it is caused by the limited responsiveness of CLIP towards directional descriptors.
> Also, we actually want to reference to the fourth image in Figuire B.13 to show that the generalization from person's upper/lower arm to person's arm is still challenging, despite the presence of the shared term "arm" . Hence, the directional terms may inadvertently impede language-driven generalization.
>
> Q5. By saying “Table 3 serves as an ablation study for ZSseg+ rather than presenting the results under the zero-shot setting”, we actually mean that Table 3 serves a different purpose with Table 1 and Table 2. We report the one-shot performance of ZSseg+ because tuning the CPTCoOp has a very low training cost as shown in Table A.1.  It may be helpful for practical use which is outside the scope of OV-PARTS benchmark.
> To avoid confusion, we have removed this part. We are appreciative of your guidance in this regard.
>
> Q6: We have adjusted  Figure 1 in the updated paper. We hope this version offers enhanced clarity and welcome any additional suggestions or corrections.

---

> > ### Comment · Reviewer_cNC2 · 2023-08-29
> >
> > Thanks for your response, the sum of changes really improves the quality and clarity of the paper and I have therefore decided to increase my rating to 7.

---

### Official Review · Reviewer_osrm · 2023-07-27
**The article presents two part-level OV dataset and three specific tasks, but additional visual analysis is still needed. The code and the data are proprietary.**

**Rating:** 6
**Confidence:** 4
**Correctness:** Yes
**Clarity:** Yes

**Strengths:**

1. The writing in the manuscript is clear and understandable.
2. The experiments are set up in a comprehensive manner, offering a complete benchmark for comparison.
3. Some improvement strategies suggested by the authors appear to be effective based on the presented data.

**Additional Feedback:**

N/A

**Documentation:**

No. For datasets, there are details on data collection and organization, availability and maintenance, and ethical and responsible use. However, URLs where reviewers can access the datasets and plans for hosting, licensing, and maintenance are not provided. For benchmarks, there is sufficient detail to support reproducibility.

**Limitations:**

Yes.

**Opportunities For Improvement:**

1. The URL is not included in the submission and the state in Checklist that "The code and the data are proprietary". This is contrary to the official requirements:
"Submission introducing new datasets must include the following in the supplementary materials (as a separate PDF):
...URL to website/platform where the dataset/benchmark can be viewed and downloaded by the reviewers.
Author statement that they bear all responsibility in case of violation of rights, etc., and confirmation of the data license.
Hosting, licensing, and maintenance plan. The choice of hosting platform is yours, as long as you ensure access to the data (possibly through a curated interface) and will provide the necessary maintenance."

2. It would be beneficial to incorporate more visual analysis into the paper. For instance, similar to the class activation maps presented in Figure 1, it would be interesting to see whether the new strategies could focus on parts such as the "sheep's head".

3. An analysis regarding the necessity of using the class+part combination in OV would be useful. What are the reasons for not just using the part? Could the word "head" be considered information leakage? Why can 'dog's head' be seen as a new category compared to 'bird's head'?

4. Could you provide a comparison between the quantitative results shown in Figure 4 and the ground truth? This might strengthen the credibility of the results and provide a clearer view of the performance of the proposed strategies.

**Relation To Prior Work:**

Yes

**Summary And Contributions:**

The authors propose an Open-Vocabulary Part Segmentation (OV-PARTS) benchmark, which includes two versions of existing datasets, namely Pascal-Part-116 and ADE20K-Part-234. This benchmark introduces three specific tasks - Generalized Zero-Shot Part Segmentation, Cross-Dataset Part Segmentation, and Few-Shot Part Segmentation. These tasks aim to provide a deeper understanding of the OV-PARTS models' analogical reasoning capabilities, their handling of open granularity, and their few-shot adapting abilities. In addition, the authors examine the two prevalent paradigms of object-level OVSS methods, and suggest how these can be adapted effectively for OV-PARTS.

---

> ### Author Response · Authors · 2023-08-18
>
> ### Q1. The release of datasets and code.
>
> Please note that we **DIDN'T** state **"The code and the data are proprietary"** in the checklist.
> The content in L418-L421 is the official examples provided to illustrate the type of statements typically found in the checklist.
> We actually don't introduce new datasets and thus give details about how we process the original Pascal-Part and ADE20K-Part in the paper and the supplemental material, which are publicly available.  We have released the code for loading the Pascal-Part-116 and ADE20K-Part-234 datasets, as well as all baselines' implementation, and checkpoints at this link: https://github.com/kellyiss/OV_PARTS .
>
> ### Q2. The class activation map after applying the strategies.
>
> The class activation map in Figure 1 aims to explain the performance gap between CLIP's zero-shot classification ability on objects and parts.  Hence, due to this inherent limitation of CLIP, part-level OVSS is more challenging than object-level OVSS. The baselines are designed to improve the existing object-level OVSS methods for parts. It's less meaningful to visualize CLIP's class activation map again since the pretrained weights are still frozen in these baselines. We mainly finetune an extra segmentation head and some lightweight modules like learnable prompts or clip adapters, which are appended to the frozen CLIP model.
>
> ### Q3. Why use class+part prompt in OV-PARTS.
>
> Part can exhibit ambiguity without object context. For example, "head" could refer to various concepts in the open world, such as "dog head," "person head," "train head," "hammer head" etc.
>
> With limited data, models trained to align the diverse visual appearance features with the same language feature will suffer from drastic intra-class variations and are apt to underfit.
> In fact, incorporating object-part relationships into the model has exhibited effectiveness  in open-world class-agnostic part segmentation[1].
>
> Our experiments also show the effectiveness of introducing object awareness to models.
> In the two-stage baselines, as shown in Table 3, the original ZSseg baseline only aligns the part feature with the object+part prompt. We further train this baseline using part prompt which shows minimal impact:
>
> | Model | Prompt | Seen | Unseen | Harmonic |
> |-------|--------|------|--------|----------|
> | ZSseg | Obj+Part | 49.35 | 12.57 | 19.76 |
> | ZSseg | Part | 49.21 | 12.68 | 20.16 |
>
> However, our proposed Object Mask Prompt technique aligns the object+part features with the object+part prompt and shows better generalizability to unseen parts.
>
> For the one-stage baselines, CLIPSeg and CATSeg trained on object datasets show certain general generalizability to part. This observation also proves the significance of aligning at the object level.
>
> ### Q4. Why can 'dog's head' be seen as a new category compared to 'bird's head'?
>
> "dog's head" seems less novel compared to "bird's head" due to the presence of "cat" while there isn't a relevant base class to "bird".
> However, given that "dog" is novel at the object level, it's appropriate to view "dog's head" as a novel class as well. Notably, in the traditional generalized zero-shot setting, significant efforts are made to avoid confusingly recognizing the novel "dog" as "cat" through semantic representations.
> Furthermore, in our OV-PARTS benchmark, we expect that the models possess genuine analogical reasoning ability, in order to generalize effectively across a diverse range of objects in the open world.
>
> Hence it's vital for the models to segment "dog's head" based on a comprehensive understanding of how "dog" relates to and differs from "cat".
> If only segmenting the "head", without the corresponding language features of "cat" and "dog", the model will inevitably overfit and treat "dog" as "cat" to parse "head".  While the performance will not drop on this class, it detracts from the objectives of our OV-PARTS benchmark.
>
> ### Q5. Ground truth in quantitative results.
>
> Thanks for the helpful advice. We have modified the qualitative results by providing the ground truth.
> Due to limited length of the paper, we reorganize the original Figure 4 and put it into the supplement material.
> Please check Figure B.5 and Figure B.7 in the supplement material.
>
> [1] Pan, T., Liu, Q., Chao, W., & Price, B.L. Towards Open-World Segmentation of Parts. Proceedings of the IEEE/CVF Conference on Computer Vision and Pattern Recognition 2023.

---

### Official Review · Reviewer_cZ7P · 2023-07-28

**Rating:** 6
**Confidence:** 4
**Correctness:** yes, the claims are correct
**Clarity:** yes, the paper is well written

**Strengths:**

1. An important benchmark for studying open vocabulary part segmentation
2. The benchmark can be used for a couple of tasks
3. Experimental analysis is provided

**Additional Feedback:**

please see the above comments

**Documentation:**

the dataset is not available and i could not evaluate this aspect

**Limitations:**

yes

**Opportunities For Improvement:**

1. the benchmark doesn't seem to be available during the review period. this is difficult for reviewers to examine the dataset.
2. do the authors intend to release code for the baseline methods considered in this paper?

**Relation To Prior Work:**

yes, it's clearly discussed

**Summary And Contributions:**

This paper proposes a benchmark for studying the problem of open vocabulary part segmentation. This problem has been made on object-level, but not part-level. Part segmentation is inherently more complicated and this benchmark introduces a new challenge in the open world due to the diverse and ambiguous definitions. The benchmark can be used to study three tasks, i.e., generalized zero-shot part segmentation, cross-dataset part segmentation and few-shot part segmentation. These are all important yet challenging tasks. The authors examine existing methods on the benchmark and provide experimental analysis.

---

> ### Author Response · Authors · 2023-08-18
>
> ### Q1. Dataset and baseline release.
>
> We actually just apply simple filtering and merging rules (explained in the paper and supplement material) to the publicly available datasets Pascal-Part and ADE20K-Part. We have released the code for loading the Pascal-Part-116 and ADE20K-Part-234 datasets, as well as all baselines' implementation, and checkpoints at this link: https://github.com/kellyiss/OV_PARTS .

---

> > ### Comment · Reviewer_cZ7P · 2023-08-24
> >
> > Thanks for releasing all baseline implementations and checkpoints. I'd like to upgrade my score.

---

### Official Review · Reviewer_r29M · 2023-07-29
**OV-PARTS: Towards Open-Vocabulary Part Segmentation**

**Rating:** 6
**Confidence:** 4
**Correctness:** Yes.
**Clarity:** Yes.

**Strengths:**

To address three challenges of open-vocabulary part segmentation, this paper proposes a benchmark containing three specific tasks and reorganized two datasets. The motivation for this paper is well justified, and implemented. The overall organization of the paper flows well.

**Additional Feedback:**

Nothing yet.

**Documentation:**

There is sufficient detail on data collection and organization.

**Ethics:**

I do not suspect any ethical concerns with the submission.

**Limitations:**

This paper provides insights into the strengths and limitations of current approaches, highlighting the potential of leveraging large foundation models for OV-PARTS.
Why select ZSseg and CLIPSeg as baseline may require to be explained, and more diverse explorations on other state-of-the-art methods are expected.

**Opportunities For Improvement:**

1. More technical details about the one-stage baseline may be expected.
2. In Lines 262-263, this paper states that CLIPSeg even outperforms 262 ZSseg+ with a ResNet-50 backbone on the unseen classes. It may need more explanation, to be consistent with the results displayed in the Table 1 and Table 2.
3. The order of table placement may need to be rearranged. In the current manuscript, Table 5 is referred before Table 4.
4. This paper requires more proofreading.


**Relation To Prior Work:**

Yes.

**Summary And Contributions:**

This paper introduces challenges of open-vocabulary part segmentation, containing the limited availability of labeled data, the complexity of part structures and the open granularity challenge, and proposes an OV-PARTS benchmark that includes two new versions of publicly available datasets.

---

> ### Author Response · Authors · 2023-08-18
>
> ### Q1. More technical details about the one-stage baselines.
>
> We added more technical details about our modifications to the original CLIPSeg for efficient finetuning in supplement material L27 - L35. For CATSeg, we adopt exactly the same architecture as in their original paper.
> The implementation of all the baselines are available here: https://github.com/kellyiss/OV_PARTS .
>
> ### Q2. Analysis of why CLIPSeg outperforms ZSseg+.
>
> Two-stage and one-stage baselines both have pros and cons as stated in L67-L71 and L78-80.
> ZSseg+ trained a MaskFormer from scratch on the training set, which results in class-agnostic part proposals that are naturally biased towards the seen classes.
> As a result, although ZSseg+ can perform well on some novel classes that bear strong resemblances to the seen classes, it fails in classes with very different parts from seen ones.  Also, ZSseg+ can't handle diverse part definitions as its part proposals are class-agnostic.
>
> On the other hand, CLIPSeg, trained with a frozen CLIP, directly decodes pixel-level visual features from its visual encoder, aligning them with the text feature. This strategy empowers the one-stage baselines with better generalization ability, outperforming the use of CLIP for independent class-agnostic proposal classification.
>
> The results in Table 1 and Table 2 confirm the characteristics of these two paradigms:
> ZSseg+ tends to overfit the training set, leading to better performance on seen classes.
> CLIPSeg demonstrates superior generalization ability, leading to performance gains on unseen classes.
>
> In the generalized zero-shot setting, harmonic mean-IoU serves as the representative metric which can punish model overfitting.
> Therefore, the results in Table 1 and 2 are consistent with the conclusion that CLIPSeg outperforming ZSseg+.
>
> ### Q3. Why select ZSseg and CLIPSeg as baselines?
>
> We choose ZSseg[1] because it is a state-of-the-art and representative two-stage method. ZegFormer[2] is a concurrent work that is the same as ZSseg. Another concurrent work OpenSeg[3] adopts the same architecture as ZSseg with a different caption-based training criterion which is not applicable to OV-PARTS.  More two-stage methods like ODISE[4], Side-Adapter[5], and OVSeg[6] are all based on the decoupling architecture of ZSseg. Our proposed improvements also apply to these works. But we would like to clarify that they fall outside the scope of this conference. Moreover, we have explored prompt tuning methods like CoOp[7] and CoCoOp[8] to ZSseg. For one-stage methods, there are other two methods: LSeg[9] and MaskCLIP[10]. LSeg can't generalize to part because it just uses the text encoder of CLIP. MaskCLIP is not trainable, thus not applicable. CATSeg[11] actually falls outside the scope of this conference. We are interested because this method exhibits zero-shot part segmentation ability.
>
> [1] Xu, Mengde, et al. "A simple baseline for open-vocabulary semantic segmentation with pre-trained vision-language model." European Conference on Computer Vision 2022.
>
> [2] Ding, Jian, et al. "Decoupling zero-shot semantic segmentation." Proceedings of the IEEE/CVF Conference on Computer Vision and Pattern Recognition. 2022.
>
> [3] Ghiasi, Golnaz, et al. "Scaling open-vocabulary image segmentation with image-level labels." European Conference on Computer Vision 2022.
>
> [4] Xu, Jiarui, et al. "Open-vocabulary panoptic segmentation with text-to-image diffusion models." Proceedings of the IEEE/CVF Conference on Computer Vision and Pattern Recognition. 2023.
>
> [5] Xu, Mengde, et al. "Side adapter network for open-vocabulary semantic segmentation." Proceedings of the IEEE/CVF Conference on Computer Vision and Pattern Recognition. 2023.
>
> [6] Liang, Feng, et al. "Open-vocabulary semantic segmentation with mask-adapted clip." Proceedings of the IEEE/CVF Conference on Computer Vision and Pattern Recognition. 2023.
>
> [7] Zhou, Kaiyang, et al. "Learning to prompt for vision-language models." International Journal of Computer Vision 130.9 (2022): 2337-2348.
>
> [8] Zhou, Kaiyang, et al. "Conditional prompt learning for vision-language models." Proceedings of the IEEE/CVF Conference on Computer Vision and Pattern Recognition. 2022.
>
> [9] Li, Boyi et al. “Language-driven Semantic Segmentation.” International Conference on Learning Representations 2022.
>
> [10] Dong, Xiaoyi, et al. "Maskclip: Masked self-distillation advances contrastive language-image pretraining." Proceedings of the IEEE/CVF Conference on Computer Vision and Pattern Recognition. 2023
>
> [11] Cho, Seokju, et al. "CAT-Seg: Cost Aggregation for Open-Vocabulary Semantic Segmentation." arXiv preprint arXiv:2303.11797 (2023).

---

### Official Review · Reviewer_NFDP · 2023-08-02
**Helpful benchmark, thorough experiments, interesting findings, some parts need further discussion and clarity**

**Rating:** 6
**Confidence:** 4

**Strengths:**

- Explored open-vocabulary part segmentation, a significant task in the research community.
- Thorough analysis of failure modes in state-of-the-art (SOTA) models, providing insights for future performance improvements.
- The datasets are the cleaned up versions of existing noisy datasets that improves evaluation quality
- Interesting finding that zero-shot performance of the OVSS models is already good on OV-PARTS
- The experiments and observations in the paper help explain the substantial performance gap in performance between OVSS task and OV-PARTS.

**Additional Feedback:**

- VLPart[1] came out recently and was discussed in Related Work section. I'm curious to know how strong VLPart performance is on this benchmark.

[1] Sun, P., Chen, S., Zhu, C., Xiao, F., Luo, P., Xie, S., & Yan, Z. (2023). Going Denser with Open-Vocabulary Part Segmentation. arXiv preprint arXiv:2305.11173.

**Clarity:**

- The paper read well, with some minor typos (e.g. "Settig" in table 5)
- Some of the tables' captions can be improved to be more self-contained.

**Correctness:**

- Claims made in the submission are correct.
- Dataset construction makes sense.
- Evaluation methods and experiment design are appropriate, supporting the claims of the paper.

**Documentation:**

- I can't find a way to access these updated datasets/annotations.
- Model hyperparameters and training details were discussed.

**Ethics:**

- Paper followed ethics guidelines.

**Limitations:**

- Limitations of the current models were mentioned in the texts. However, the limitations of the benchmark were not discussed. Is there any aspect of open-vocabulary part segmentation that the proposed tasks did not cover? What can potentially be further improved on the data end in the future?
- Potential negative societal impact was not discussed in either the main text or the supplement.

**Opportunities For Improvement:**

- Unclear explanation of blank slots in Tables 1 and 2, particularly regarding the presence or absence of numbers in the Pred obj columns.
- Some terminologies were not clearly defined. What exactly is "mask denoise" strategy used for ZSeg?
- Ambiguity in how models were evaluated on Pascal Part during cross-category experiments. It's unclear if the text prompts were based on Pascal Part or ADE20K vocabulary.
- Suggestion to explore training models on Pascal Part and testing on ADE20K to understand potential failures in fine-grained part boundaries or language understanding. It would be interesting to see if the failures of these models in this case (if they fail) were due to not understanding on the language side, i.e. unable to guess that "bumper" is in the front of the car, or not knowing the fine-grained part boundaries.
- What is the necessity of inputting text prompt as "cow's leg" instead of "leg" since models seem to focus on a single object in the scene?
- In the Supplement results shown on ADE2K data where there are multiple objects in the scene seem to focus on only 1 object in the scene (e.g. scene with multiple chairs only show part prediction on 1 chair). Discussion about this would be helpful.



**Relation To Prior Work:**

- The paper discussed the differences between their contributions and other related works, including open-vocabulary semantic segmentation, open-world part segmentation and open-vocabulary part segmentation.
- There are many works that tackle the task of Open-vocabulary part detection (e.g. GLIP, VLDet, etc). Please discuss the connections between these task and OV-Parts?

**Summary And Contributions:**

- The paper introduced Open-Vocabulary Part Segmentation benchmark with a focus on zero-shot, cross-category, and few-shot part segmentation tasks.
- The benchmark evaluations were conducted on two updated datasets: ADE20K Part and Pascal Part.
- The paper analyzed foundation models, classifying them into one-stage and two-stage methods, with extensive experiments performed to assess models performance in OV-PARTS.
- The paper shed lights onto the potential of using/tuning vision language foundation models to perform part segmentation.
- The paper introduced strategies to improve future part segmentation models based on observations about the drawbacks of the pretrained foundation models.

---

> ### Author Response · Authors · 2023-08-18
>
> ### Q1. Explanation of blank slots in Tables 1 and 2.
>
> For the supervised reference MaskFormer, the harmonic-IoU metric, which is designed for the zero-shot setting, is intentionally ignored.
>
> For the two-stage baselines in Table 1 and Table 2,  the omission of ZSseg+/ResNet-50/CPTCoCoOp under Pred-Obj is intentional, as it was designed as an ablation study on CPTCoOp and CPTCoCoOP. To simplify the analysis, the results are only analyzed under Oracle-Obj, as both models use the same object model.
>
> Furthermore, in Table 2, we apologize for the missing result for ZSseg+/ResNet-101c/CPTCoOp and have added it to the revised version.
> For the one-stage baselines, their design enables them to generalize directly to both object-level and part-level without the need for additional object models. The reported results for CATSeg and CLIPSeg without finetuning under the Oracle-Obj setting serve the purpose of showing their impressive zero-shot part performance.
> In fact, it's not a fair comparison with other baselines that are finetuned on the part datasets.
> Hence, we choose to skip the results under the Pred-Obj setting.
>
> ### Q2. Mask denoise strategy.
>
> Due to the relatively lower quality of part proposals in comparison to object proposals, we take a different approach for CLIP classification.
>
> Firstly, we compute the class-aware segmentation masks, exhibiting a bias to the seen objects, as in the original Maskformer. Then we binarize it and relabel each binary mask by sending it to CLIP and then fuse the two score maps to obtain the final result. We have modified the formula to make it clear in *L208-L212*.
>
> This practice is similar to VLPart which uses DINO features to find the most similar seen object for each unseen object and parses the parts by imitating the seen parts. It works well when we annotate objects with conformed parts (*e.g.*, dog and cat, car and bus share the same part set in Pascal-Part-116), but it would be harder to generalize to unseen classes that don't have corresponding base classes. This also explains why two-stage baselines perform worse on the unseen classes compared to the one-stage baselines.
>
> ### Q3. Ambiguity in cross-dataset evaluation.
>
> The text prompts are always based on the target dataset under the cross-dataset setting. If not, we won't be able to calculate the metrics on the target dataset.
>
> ### Q4. Pascal-Part-116 to ADE20K-Part-234 evaluation.
>
> Thanks for the constructive suggestion. We have provided more qualitative results and analysis of this reversed evaluation as shown in Figure B.13 (supplement material).
>
> ### Q5. Text prompt as "cow's leg" instead of "leg".
>
> Part can exhibit ambiguity without object context. For example, "leg" could refer to various concepts in the open world, such as "cow's leg", "person's leg" "table's leg" etc.
>
> With limited data, models trained to align the diverse visual appearance features with the same language feature will suffer from drastic intra-class variations and are apt to underfit.
> In fact, incorporating object-part relationships into the model has exhibited effectiveness in open-world class-agnostic part segmentation[1].
>
> Our experiments also show the effectiveness of introducing object awareness to models.
> In the two-stage baselines, as shown in Table 3, the original ZSseg baseline only aligns the part feature with the object+part prompt. We further train this baseline using part prompt which shows minimal impact:
>
> | Model | Prompt | Seen | Unseen | Harmonic |
> |-------|--------|------|--------|----------|
> | ZSseg | Obj+Part | 49.35 | 12.57 | 19.76 |
> | ZSseg | Part | 49.21 | 12.68 | 20.16 |
>
> However, our proposed Object Mask Prompt technique aligns the object+part features with the object+part prompt and shows better generalizability to unseen parts.
>
> For the one-stage baselines, CLIPSeg and CATSeg trained on object datasets show certain general generalizability to part. This observation also proves the significance of aligning at the object level.
>
> ### Q6. Only one object instance has the results.
>
> ADE20K annotates parts at the object instance level, but not all object instances are equally annotated.
> Hence, we implement the evaluator based on object instances with parts and give visualizations per object instance. Sorry about the confusion and actually the model predicts part labels for all pixels.
>
> ### Q7. Limitations of the benchmark concerning open-vocabulary part segmentation and potential negative societal impact.
>
> We have added limitations and potential negative societal impacts in Section D (supplement material).
>
> [1] Pan, T., Liu, Q., Chao, W., & Price, B.L. (2023). Towards Open-World Segmentation of Parts. Proceedings of the IEEE/CVF Conference on Computer Vision and Pattern Recognition 2023.

---

> ### Author Response · Authors · 2023-08-18
> **Relation To Prior Work.**
>
> ### Q1. Connections with other open-vocabulary detection models.
>
> Some existing open-vocabulary object detection models trained on crowd-sourced datasets can also detect parts with bounding boxes since the texts can include part words. However, pixel-level understanding is more conforming to the nature of the part which is ambiguous, multi-granular and has intricate boundaries. For example, with bounding boxes, it'll be hard to identify the little bird's parts as shown in supplement Figure B.5 or achieve the multi-granular generalization as shown in Figure B.6.
>
> ### Q2. VLPart on OVPart benchmark.
>
> VLPart is also a part detection model that does not provide pixel-level parsing, making it unsuitable to be directly evaluated using the same metrics as our benchmark.

---

> ### Comment · Reviewer_NFDP · 2023-08-24
> **Additional Comments**
>
> I would like to thank the authors for their responses to my concerns and questions. While most of my questions and concerns were addressed, I have the following suggestions/comments:
> 1. For completeness I would encourage the authors to fill in the missing slots in Tables 1 and 2 as best as they can and clearly explain in the paper the reason why they omit some numbers as it was confusing for me.
> 2. As far as I know VLPart provides pixel-level mask prediction and not part detection. I acknowledge that VLPart came out just recently so the comparison of VLPart on this benchmark is not needed. However I still think that it would be interesting and helpful to the community to see how VLPart performs on the benchmark.
>
> I'm keeping my original rating. The authors can feel free to incorporate my follow up feedback in the final version.

---

> ### Author Response · Authors · 2023-08-28
>
> ### Q1. Missing slots in Tables 1 and 2.
>
> We have complemented all the missing slots in Table 1 and Table 2. We are appreciative of your guidance in this regard.
>
> ### Q2. Results of VLPart on OV-PARTS.
>
> We're sorry about not making it clear. Although VLPart can output masks, they actually use Mask R-CNN to do instance segmentation for parts which should be based on detection boxes. As a result, they adopt a different approach for processing the part data. Take the Pascal-Part as an example, which is a common dataset used by OV-PARTS and VLPart, VLPart need to convert the part mask annotations to part boxes using the minimum bounding box of the mask as in https://github.com/facebookresearch/VLPart/blob/main/tools/pascal_part_mat2json.py#L192C22-L192C22 .
>
> This causes two differences with our Pascal-Part-116.
> 1. A pixel can have multiple labels. For example, in VLPart, the part mask/box of person's head contains the masks/boxes of person's eyes, person's nose, person's mouth etc. While in OV-PARTS, each pixel has only one label.
>
> 2. Since fine-grained parts of an object usually congregate closely and have intricate boudaries, minimum bounding boxes would be inaccurate. We find that VLPart actually discards some parts of objects like "aeroplane's engine", "bicycle's chainwheel" etc., resulting less part classes (93 v.s 116), as shown in https://github.com/facebookresearch/VLPart/blob/main/vlpart/data/datasets/pascal_part.py
>
> If applying VLPart to OV-PARTS, firstly we need to convert Pascal-Part-116 to instance-level part segmentation data which may cause inconsistent and inaccurate box annotations. Then we need to convert the results back to semantic segmentation masks which requires various postprocessing to assign each pixel a part label. Hence, we still think it's unsuitable to be directly evaluated in OV-PARTS.
>
> We agree that the way they build dense semantic correspondence between the novel and base classes using DINO features can be applicable in OV-PARTS. But we'll need more time to transfer this part into our zero-shot semantic segmentation pipeline. Moreover, as we have mentioned, relating novel classes to base classes has limitations in generalizing to more novel parts. We note that VLPart only sets two novel object classes "bus" and "dog" which share the same part set as "car" and "dog" as shown in https://github.com/facebookresearch/VLPart/blob/main/vlpart/data/datasets/pascal_part.py#L186C6-L186C6 .

---

### Author Response · Authors · 2023-08-18

We express our sincere gratitude to all the reviewers for their insightful feedback. We are greatly encouraged by the generally positive reception and recognition of the strengths in OV-PARTS,
(1) A Novel and Important Benchmark: The paper well justifies the importance and challenges of open vocabulary part segmentation.
(2) Significant Benchmark Setup: The introduction of three specific tasks within OV-PARTS demonstrates careful consideration and practical relevance.
(3) Enhanced Datasets: The refinement of existing datasets into cleaner versions, Pascal-Part-116 and ADE20K-Part-234, enhances the quality of evaluation.
(4) Comprehensive Experimental study: The proposed baselines yield effective results and provide deep insights into potential improvements for future works with foundation models.
(5) Good Writing.

Furthermore, the major concern of **Reviewer cZ7P** and **Reviewer osrm** is about code and data release.
We **DIDN'T** state **"the code and the data are proprietary"** in the checklist and  it is our utmost priority to ensure that OV-PARTS adhere to the criteria of this track.
Since the datasets are publicly available and accessible, apart from giving each of the details about how we filter and merge the original data in the paper and supplement material, we have released the code for loading the Pascal-Part-116 and ADE20K-Part-234 datasets, as well as all baselines' implementation, and checkpoints at this link: https://github.com/kellyiss/OV_PARTS .
Moreover, we have revised both the paper and the supplementary material according to the feedback. The modified content has been highlighted in cyan for easy identification and reference.

---

### Decision · Program_Chairs · 2023-09-22

**Decision:**

Accept (Poster)

**Comment:**

The paper receives borderline reviews. But importantly, all 7 reviewers were leaning towards acceptance, and no major concerns with the dataset were raised post-rebuttal. Key elements for dataset and benchmarking papers, including formulation of the evaluation (r29M, cZ7P, v7gi, cNC2), the benchmarking and analysis of SOTA failures (NFDP, cZ7P, osrm, v7gi, VdPY, cNC2) and the insights provided by the work (NFDP, osrm, cNC2), are all presented properly in the work. Therefore, AC recommends for acceptance.